

# Decomposition aided attention-based recurrent neural networks for multistep ahead time-series forecasting of renewable power generation

Robertas Damaševičius[1], Luka Jovanovic[2], Aleksandar Petrovic[3], Miodrag Zivkovic[3], Nebojsa Bacanin[3], Dejan Jovanovic[4] and Milos Antonijevic[3]

[1] Department of Applied Informatics, Vytautas Magnus University, Kaunas, Lithuania
[2] Faculty of Technical Sciences, Singidunum University, Belgrade, Serbia
[3] Faculty of Informatics and Computing, Singidunum University, Belgrade, Serbia
[4] College of Academic Studies "Dositej", Belgrade, Serbia

Corresponding author
Robertas Damaševičius,
robertas.damasevicius@vdu.lt

## ABSTRACT

Renewable energy plays an increasingly important role in our future. As fossil fuels become more difficult to extract and effectively process, renewables offer a solution to the ever-increasing energy demands of the world. However, the shift toward renewable energy is not without challenges. While fossil fuels offer a more reliable means of energy storage that can be converted into usable energy, renewables are more dependent on external factors used for generation. Efficient storage of renewables is more difficult often relying on batteries that have a limited number of charge cycles. A robust and efficient system for forecasting power generation from renewable sources can help alleviate some of the difficulties associated with the transition toward renewable energy. Therefore, this study proposes an attention-based recurrent neural network approach for forecasting power generated from renewable sources. To help networks make more accurate forecasts, decomposition techniques utilized applied the time series, and a modified metaheuristic is introduced to optimized hyperparameter values of the utilized networks. This approach has been tested on two real-world renewable energy datasets covering both solar and wind farms. The models generated by the introduced metaheuristics were compared with those produced by other state-of-the-art optimizers in terms of standard regression metrics and statistical analysis. Finally, the best-performing model was interpreted using SHapley Additive exPlanations.

## INTRODUCTION

The role of renewable energy is a paramount factor in sustainability of the society. Traditional energy systems based on fossil fuels are not efficient and require more complicated processes of extraction. The demands of human civilization are always growing, which exposes the difficulties for eco-friendly energetic growth. As renewable

energy source (RES) become more available the distribution of new resources in the network result in stochasticity, intermittency, and uncertainty. Consequentially, the traditional energy systems are dominant in the share of energy used amounting to 81% of the global share (*Loe, 2022*).

For RES to become more widely utilized, the previously mentioned challenges need to be overcome. Additionally, energy storage on a smaller scale remains difficult when working with RES, in comparison to fossil fuel storage which is still considered more reliable. The storage of electricity is mostly achieved by batteries which are a limited resource on their own due to the limited number of life cycles for each one of them (*Zhang & Zhao, 2023*). All things considered, a possible solution is a mechanism that can provide accurate forecasts of the amount of resources being generated from RES. Such a solution would have to be able to analyze short-term time series and provide a robust mechanism as it affects electricity load and its price. Electricity traders and system operators are most affected by these changes.

Traditional methods for regression have previously been applied to forecasting RES power production (*Foley et al., 2012*; *Abuella & Chowdhury, 2015*) However, as the world's need for energy increases further improvements are needed in order to make forecasting methods viable. A major challenge when tackling RES production forecasting comes from the noisy nature of the data. Since renewable resources rely on natural phenomena such as wind or solar exposure, many chaotic factors play a role in the amount of power that can be produced. Nevertheless, patterns in this data are still present, though often difficult to initially observe.

By applying advanced signal processing techniques, such as decomposition techniques, strong signals can be separated from the noise, allowing prediction methods to focus on determining correlations between signals with strong patterns rather than those heavily affected by the noise. This concept has often been applied to systems that require precise moments in noise environments such as electroencephalography (*Murariu, Dorobanţu & Tărniceriu, 2023*) demonstrating great potential. Several decomposition techniques have been developed in recently such as empirical mode decomposition (EMD) (*Boudraa & Cexus, 2007*) and ensemble empirical mode decomposition (EEMD) (*Wu & Huang, 2009*). While efficient, the lack of a strong mathematical background in these methods has led to the development of variational mode decomposition (VMD) (*Dragomiretskiy & Zosso, 2013*) that has shown great potential for tackling signal decomposition with a strong mathematical basis (*Liu et al., 2022*; *Zhang, Peng & Nazir, 2022*; *Gao et al., 2022*).

One additional approach that has shown great potential when working with data catheterized by complex nonlinear relations is the application of artificial intelligence (AI). Powerful AI algorithms are capable of improving their performance through an iterative data-driven process. By observing data AI algorithms can determine correlations without explicit programming. This makes AI a promising approach for tackling this pressing issue. Nevertheless, the modern algorithms' performance is reliant on proper hyperparameter selection. With increasing numbers of hyperparameters, traditional methods such as trial and error have become insufficient to optimize algorithm

performance. The use of metaheuristic optimization algorithms provides a potential solution for efficient hyperparameter selection.

Forecasting power generation is regarded as a time series forecasting challenge. By doing so, algorithms capable of responding to data sequences can be leveraged in order to make more accurate forecasts. One promising approach, that extensive literature review suggests has not yet sufficiently been explored when applied to renewable forecasting, is the use of recurrent neural networks (RNN) (*Medsker & Jain, 1999*). These networks represent a variety of artificial neural networks (ANN) that allow previous inputs to affect future outputs, making them highly suitable for time series forecasting. A recent improvement incorporates attention mechanisms (*Olah & Carter, 2016*) into RNN allowing networks to focus their attention on specific features improving accuracy. Additionally, the literature review suggests that attention-based RNNs (RNN-ATT) have not yet been applied to renewable power forecasting, indicating a gap in research that this work hopes to address. Exploring the potential of these networks is essential as a robust forecasting method could help make RES more viable and lower the world's dependence on fossil fuels.

This research proposes an approach that applies a neural network model based on attention for that purpose. Moreover, the proposed model was applied to two different problems including the Spain wind and solar energy predictions and the wind farms in China predictions. Datasets for both countries' surveys have been used with the RNN model and the attention-based recurrent neural network RNN-ATT. However, these networks require fine-tuning of a large number of hyperparameters, that can result in non-deterministic polynomial time complexity (NP-hard). Hyperparameter optimization is done through the use of metaheuristics, and a modified version of the well-known Harris hawk optimization (HHO) (*Heidari et al., 2019*) algorithm is introduced. Two sets of experiments have been carried out both with RNN and RNN-ATT networks, applied to each real-world dataset.

This research is an extension of previous researches in this domain (*Bacanin et al., 2023c*; *Stoean et al., 2023*; *Bacanin et al., 2023b*), where the long short-term (LSTM), bidirectional LSTM (BiLSTM) and gated recurrent unit (GRU) were applied for RES forecasting challenges. However, the goal of this work is to test lighter models (classical RNNs) for problems of RES with the application of fewer neurons over layers while providing satisfactory performance. Additionally, conversely to previous experimentation, current research also investigates the potential of RNNs with attention mechanism and it was validated against different RES time-series datasets. Also, the classical RNNs (without attention mechanism) were also validated in order to establish the influence of attention layer to overall network performance.

The primary contributions of this work can be summarized as the following:

- The RNN-ATT-based method for forecasting RES power generation.
- A modified version of a metaheuristic tasked with selecting network parameters.
- The application of the introduced approach to two real-world datasets to determine their potential for real-world use.

- The interpretation of the best generated RNN models that can be used as a valuable tools for renewable energy specialists to determine which factor has the most influence on the RES performance.

The structure of the article includes "Background and Preliminaries" for providing the technological fundamentals for the performed experiments. "Proposed Method" explains the original version of the applied metaheuristic as well as the modified version. "Dataset Description and Experiments" explains the utilized datasets in detail and gives information on the test setup. The outcomes are presented in "Results and Comparison", followed by a discussion. statistical validation and model interpretation presented in "Discussion, Statistical Validation and Interpretation". Finally, "Conclusions" concluded the work and presents potential future research.

# BACKGROUND AND PRELIMINARIES

This section introduces techniques required for the reader to have a full and insightful understanding of experiments conducted in this research.

## Time-series decomposition and integration

Time-series decomposition is a technique used to break down a time-series data into its constituent components, such as trend, seasonality, and residual (noise). By decomposing a time-series, we can better understand the underlying patterns and relationships within the data, which can, in turn result in improvements of reliability and accuracy of the time-series forecasting, models like the Luong attention-based RNN model.

### Decomposition techniques

Various decomposition techniques can be applied to time-series data, including:

**1. Classical decomposition:** This method decomposes a time-series into its trend, seasonal, and residual components using moving averages and seasonal adjustments. There are two primary approaches in classical decomposition: additive and multiplicative. In the additive decomposition, the time-series is expressed as the sum of its components, while in the multiplicative decomposition, the time-series is expressed as the product of its components.

**2. Seasonal and trend decomposition using Loess (STL):** STL is a flexible and robust decomposition method that uses locally weighted regression (Loess) to estimate the trend and seasonal components of a time-series. It can handle both constant and time-varying seasonality, as well as arbitrary patterns of missing data. The STL method also allows for user-defined control over the smoothness and periodicity of the seasonal and trend components.

**3. Seasonal decomposition of time series (SDTS):** SDTS is an extension of the classical decomposition method that incorporates a seasonal adjustment factor for each observation in the time-series. This factor is obtained by dividing the observed value by the corresponding seasonal component. The seasonal adjustment factors can be used to

deseasonalize the time-series, which can then be analyzed for trend and residual components.

**4. Wavelet transform:** Wavelet transform is a mathematical technique used to decompose a time-series into a set of wavelet coefficients, which represent the time-series at different scales and resolutions. Wavelet transform can capture both the low-frequency (trend) and high-frequency (seasonal and noise) components of a time-series, making it a powerful tool for time-series decomposition and analysis.

**5. Empirical mode decomposition:** EMD is a powerful and flexible technique for analyzing non-stationary and non-linear time series data. Introduced by *Huang et al. (1998)*, EMD is designed to adaptively decompose a time series into a finite set of intrinsic mode functions (IMFs) that capture the local oscillatory behavior of the signal at various scales. The primary goal of EMD is to provide a data-driven decomposition that does not rely on any predefined basis functions or assumptions about the underlying signal characteristics (*Abayomi-Alli et al., 2020*). By incorporating EMD into the renewable power generation forecasting process, we can potentially enhance the accuracy, reliability, and interpretability of the forecasting models, ultimately aiding in the efficient management and planning of renewable energy resources.

### Variational mode decomposition

The VMD (*Dragomiretskiy & Zosso, 2013*) technique used for signal decomposition builds upon the solid foundation established but other methods. However, VMD does so with a strong mathematical foundation compared to empirical techniques. Signal modes of varying frequencies are extracted from the original signal original signals by finding modes that are orthogonal to each other with localized frequency content. The decomposition is achieved through progressive optimization according to Eq. (1).

$$E(V) = \int \left( \frac{1}{2} ||V'(t)||_2^2 + \mu U(V(t)) \right) dt \tag{1}$$

in which $V(t)$ are signal modes, $V'(t)$ denotes the derivative of $V(t)$ with respect to time. Additionally the regularization parameter $\mu$ balances between extracted mode smoothness and sparsity. Accordingly, function $U(V(t))$ promotes sparsity.

The decomposition process is handled by an algorithm that switches between solving modes and determines the penalty. Minimizing the energy function modes can be determined with respect to $V(t)$. A Lagrange multiplier $\alpha(t)$ is also introduced giving Eq. (2).

$$E(V) = \int \left( \frac{1}{2} ||V'(t)||_2^2 + \mu U(V(t)) + \alpha(t) \sum_{k=1}^{K} V_k(t)^2 \right) dt \tag{2}$$

where the $k$-th mode of a signal is represented by $V_k(t)$. In order to revise the penalty function, the energy function is minimized with respect to $\alpha(t)$. To accomplish this, the derivative of $E(V)$ with respect to $\alpha(t)$ is set to zero. The resulting function is shown in Eq. (3)

$$\frac{d}{dt}\alpha(t) = \mu \sum_{k=1}^{K} V_k(t)^2 - \lambda \tag{3}$$

with the $\lambda$ constraint defining the overall mode energy.

*Integration of decomposed components*

Once the time-series has been decomposed into its constituent components, the next step is to integrate these components into the forecasting model. There are several ways to incorporate the decomposed components into the Luong attention-based RNN model:

**1. Component-wise modeling:** Train separate RNN models for each of the decomposed components (trend, seasonal, and residual), and then combine the forecasts from these models to obtain the final forecast for the original time-series. This approach can help in capturing the unique patterns and dependencies within each component more effectively.

**2. Feature augmentation:** Use the decomposed components as additional input features to the RNN model, along with the original time-series. This approach can help the model in learning the relationships between the decomposed components and the target variable, potentially improving the model's forecasting performance.

**3. Preprocessing:** Deseasonalize the time-series by removing the seasonal component before training the RNN model, and then add back the seasonal component to the model's forecasts to obtain the final forecast for the original time-series. This approach can help in reducing the complexity of the time-series and make it easier for the model to capture the underlying trend and residual patterns.

**4. Postprocessing:** Train the RNN model on the original time-series, and then adjust the model's forecasts using the decomposed components (*e.g.*, by adding the seasonal component to the model's forecasts). This approach can help in correcting the model's forecasts for any systematic errors or biases related to the seasonal component.

## Recurrent neural network

Time series prediction is the motivation for the improvements in artificial neural networks (ANN) (*Pascanu, Mikolov & Bengio, 2013*). The difference from the multilayer perceptron is that the hidden unit links are enabled with a delay. The results of such modifications allow the model to be sensitive toward temporal data occurrences of greater length.

RNNs are considered as a high-performing solution but further improvements were applied to achieve even greater performance. The main issues are the exploding and vanishing gradient. The solution was provided with LSTM model. The reason for not using the latest solution is that sometimes RNNs tend to outperform LSTMs as they introduce a large number of hyperparameters that can sometimes hinder performance (*Bas, Egrioglu & Kolemen, 2021*).

The advantage of the RNN as well is that it does not have to take inputs of fixed vector length, in which case the output has to be fixed as well. While working with rich structures and sequences this advantage can be exploited. In other words, the model works with input vectors and is able to generate sequences on the output. The RNN processes the data of the sequence while the hidden state is held.

## Luong attention-based model

The attention phenomenon is not defined by mathematics and its application in the Luong attention-based model should be considered as a mechanism (*Luong, Pham & Manning, 2015*; *Raffel et al., 2017*; *Harvat & Martín-Guerrero, 2022*). Some examples of different mathematical expression applications of the attention mechanism are the sliding window methods, saliency detection, local image features, *etc.* Regarding the attention mechanism application in the case of an RNN, the definition is precise.

The networks that can work with the attention mechanism and possess RNN characteristics are considered attention-based. The purpose of such a mechanism is to work with different weights for the sequence in input. The data can be captured as a result and input-output relations are usable. The basic solution of such architecture is the application of a second RNN.

The authors chose the Luong attention-based model for that purpose. Weight represented as $w_t$ is calculated for the source for every timestep $t$ for the decoding of attention-based encoder-decoder as $\Sigma_s w_t(s) = 1$ and $\forall s\ w_t(s) \geq 0$. The hidden state $h_t$ has a function that is the related timestep's predicted token, while the $\Sigma_s w_t(s) * \hat{h}_s$.

Different mathematical applications of the attention mechanism differ in the way they compute weights. In the case of the Luong model, it is the softmax function on the scaled scores of each token. Matrix $W_a$ linearly transforms the decoder's $h_t$ dot product and the encoder $\hat{h}_s$ to calculate the score.

## Hyperparameters of luong-attention based RNN

The Luong attention-based RNN model is an extension of the basic RNN model with the addition of an attention mechanism allows for selective focus on particular parts of the input sequence upon output generation. The following hyperparameters are typically involved in the configuration of the Luong attention-based RNN model:

**1. Number of hidden layers ($n_{hid}$):** The number of hidden layers in the RNN architecture, which determines the depth of the model. More hidden layers can enable the model to capture patterns of higher complexity and data dependencies but with the risk of overfitting and requiring more computational resources.

**2. Number of hidden units per layer ($n_{unit}$):** The number of hidden units (neurons) in each hidden layer of the RNN. A larger number of hidden units can increase the model's capacity to learn complex patterns, but it may also increase the risk of overfitting and require more computational resources.

**3. Type of RNN cell:** The choice of RNN cell used in the model, such as LSTM or GRU. These cells are designed to better handle long-range dependencies and mitigate the vanishing gradient problem compared to the traditional RNN cells.

**4. Attention mechanism:** The specific attention mechanism used in the model. In the case of the Luong attention-based RNN model, the attention mechanism can be of two types: global or local attention. Global attention attends to all the source positions, while attention is focused locally only on a small window of source positions around the current target position.

**5. Attention scoring function:** The scoring function computes the alignment scores between the source and target sequences in the attention mechanism. *Luong, Pham & Manning (2015)* proposed three different scoring functions: dot product, general (multiplicative), and concatenation (additive). The choice of scoring function can affect the model's performance and interpretability.

**6. Learning rate ($\alpha$):** The learning rate is a critical hyperparameter in control of the size of updates to the model's weights during the training process. A smaller learning rate might lead to more precise convergence but require more training iterations, while a larger learning rate may speed up the training process but risk overshooting the optimal solution.

**7. Dropout rate ($p_{drop}$):** The dropout rate is a technique of regularization used to prevent overfitting in neural networks. During training, a fraction of the neurons in the network is randomly "dropped out" or deactivated, with the specified dropout rate determining the proportion of neurons deactivated at each training iteration.

**8. Batch size:** The number of training samples used in a single update of the model's weights. A larger batch size can lead to more accurate gradient estimates and faster training but may require more memory and computational resources.

**9. Sequence length:** The length of input and output sequences used in the model. Longer sequences may allow the model to capture more extensive temporal dependencies but can also increase the computational complexity and risk of overfitting.

These hyperparameters play a paramount role in performance determination of the Luong attention-based RNN model for renewable power generation forecasting. Selecting optimal values for these hyperparameters requires careful experimentation, and metaheuristic optimization techniques like the HHO algorithm can be helpful in this process, as shown by different authors recently (*Tayebi & El Kafhali, 2022*; *Bacanin et al., 2022a*; *Nematzadeh et al., 2022*; *Drewil & Al-Bahadili, 2022*; *Akay, Karaboga & Akay, 2022*; *Bacanin et al., 2022c*; *Jovanovic et al., 2023a*).

## Metaheuristic optimization

In recent years model optimization has become a popular topic in computer science. Increasing model complexity, as well as growing numbers of hyperparameters of modern algorithms, has made it necessary to develop techniques to automate this process, which was traditionally handled through trial and error. However, this is a challenging task, as selecting optimal parameters is often a mixed NP-hard problem, with both discrete and continuous values having a role to play in defining model performance. A powerful group of algorithms capable of addressing NP-hard problems within reasonable time constraints and with realistic computational demands are metaheuristic optimization algorithms. By formulating the process of parameter selection as an optimization task, metaheuristics can be employed to efficiently improve performance. A notably popular group of metaheuristics is swarm intelligence that models observed behaviors of cooperating groups to perform optimizations. Some notable algorithms that have become popular for tacking optimization tasks among researchers include the HHO (*Heidari et al., 2019*), genetic algorithm (GA) (*Mirjalili & Mirjalili, 2019*), particle swarm optimizer (PSO) (*Kennedy & Eberhart, 1995*), artificial bee colony (ABC) (*Karaboga, 2010*) algorithm, firefly algorithm

(FA) (*Yang & Slowik, 2020*). Additionally the LSHADE for Constrained Optimization with Levy Flights (COLSHADE) algorithm (*Gurrola-Ramos, Hernàndez-Aguirre & Dalmau-Cedeño, 2020*) and Self-Adapting Spherical Search (SASS) (*Zhao et al., 2022*) are notable recent examples of optimizers. These algorithms, and algorithms derived from their base have been applied in several fields with promising outcomes. Some noteworthy examples of metaheuristics applied to optimization problems include examples for crude oil price forecasting (*Jovanovic et al., 2022*; *Al-Qaness et al., 2022*), Ethereum and Bitcoin prices predictions (*Stankovic et al., 2022b*; *Milicevic et al., 2023*; *Petrovic et al., 2023*; *Gupta & Nalavade, 2022*), industry 4.0 (*Jovanovic et al., 2023b*; *Dobrojevic et al., 2023*; *Para, Del Ser & Nebro, 2022*), medicine (*Zivkovic et al., 2022a*; *Bezdan et al., 2022*; *Budimirovic et al., 2022*; *Stankovic et al., 2022a*), security (*Zivkovic et al., 2022b*; *Savanović et al., 2023*; *Jovanovic et al., 2023c*; *Zivkovic et al., 2022c*), cloud computing (*Thakur & Goraya, 2022*; *Mirmohseni, Tang & Javadpour, 2022*; *Bacanin et al., 2022d*; *Zivkovic et al., 2021*), and environmental sciences (*Jovanovic et al., 2023d*; *Bacanin et al., 2022b*; *Kiani et al., 2022*).

# PROPOSED METHOD

This section begins with a short overview of the basic HHO algorithm along the explanation and justifications of the modifications that were made to the original method.

## Original Harris hawk optimization

The inspiration for the HHO are the attack strategies of the bird with the same name. The phases of attacks can be differentiated as exploration, the transition to exploitation, and the exploitation. The algorithm was introduced by *Heidari et al. (2019)* and has been used for a wide variety of optimization-related applications such as machine scheduling (*Jouhari et al., 2020*) and neural network optimization (*Ali et al., 2022*).

In the first phase, the exploration, the goal is the global optimum. Multiple locations in the population serve for random initialization which mimics the hawk's search for prey. The parameter $q$ controls this process as it switches between two strategies of equal probability:

$$X(t+1) = \begin{cases} X_{rand}(t) - r_1|X_{rand}(t) - 2r_2X(t)|, q \geq 0 \\ (X_{best}(t) - X_m(t)) - r_3(LB + r_4(UB - LB)), q < 0.5, \end{cases} \quad (4)$$

in which the random number from the range $[0, 1]$ are $r_1$, $r_2$, $r_3$, and $r_4$ as well as $q$ and these numbers are updated on an iteration basis. The position vector of the solution in the next iteration is $X(t+1)$, and the positions of the solutions of the best, current, and average solutions in the current iteration $t$ are given respectively as $X_{best}(t)$, $X(t)$ and $X_m(t)$, while the lower bound is $LB$ and the upper bound is $UB$. The average position is provided by a simple averaging approach:

$$X_m(t) = \frac{1}{N}\sum_{i=1}^{N} X_i(t), \quad (5)$$

for which $N$ shows the total solutions number, and the individual $X$ at iteration $t$ is shown as $X_i(t)$.

The term prey energy is introduced as it indicates if the algorithm should revert back to exploration and so forth. The solutions updates strength in each iteration as:

$$E = 2E_0\left(1 - \frac{t}{T}\right), \tag{6}$$

for $T$ as iteration maximum for a run, the prey's initial energy $E_0$ which varies inside the $[-1, 1]$ interval.

The exploitation phase represents the literal attack of the hawk and maps out its behavior as it is closing in. The mathematical translation is given as $|E| \geq 0.5$ for more passive attacking, and $|E| < 0.5$ otherwise.

In cases where the prey of the hawk is still at large, the hawks encircle the prey with the goal of exhaustion which is modeled as follows:

$$X(t+1) = \Delta X(t) - E|JX_{best}(t) - X(t)| \tag{7}$$
$$\Delta X(t) = X_{best}(t) - X(t), \tag{8}$$

for which the vector difference of the best solution (prey) and the current solution in iteration $t$ is shown as $\Delta X(t)$. The strategy of the prey's escape is controlled by the random attribute $J$ which differs from iteration to iteration:

$$J = 2(1 - r_5), \tag{9}$$

for which the interval $[0, 1]$) maps out the random value $r_5$. For $r \geq 0.5$ and $|E| < 0.5$ the prey is considered exhausted and more aggressive attack strategies are applied. The current position in this case is updated as:

$$X(t+1) = X_{best}(t) - E|\Delta X(t)| \tag{10}$$

If the prey is still not giving up the hawks apply another attack strategy called zig-zag movements commonly known as leapfrog movements. Following equation evaluates if such behavior should be applied:

$$Y = X_{best}(t) - E|JX_{best}(t) - X(t)|, \tag{11}$$

while the leapfrog movements are modeled as:

$$Z = Y + S \times LF(D), \tag{12}$$

in which the problem dimension is given as $D$, a random vector of $1 \times D$ size as $S$, and the levy fligth $LF$ calculated by:

$$LF(x) = 0.01 \times \frac{u \times \sigma}{|v|^{\frac{1}{\beta}}}, \sigma = \left(\frac{\Gamma(1 + \beta) \times sin(\frac{\pi\beta}{2})}{\Gamma(\frac{1+\beta}{2}) \times \beta \times 2^{(\frac{\beta-1}{2})}}\right)^{\frac{1}{\beta}} \tag{13}$$

Consequently, the position updating mechanism is provided:

$$X(t+1) = \begin{cases} Y, & \text{if } F(Y) < F(X(t)) \\ Z, & \text{if } F(Z) < F(X(t)), \end{cases} \tag{14}$$

where the Eqs. (11) and (12) are utilized for calculating the $Y$ and $Z$.

Lastly, for the case of $r \leq 0.5$ and $|E| < 0.5$ the prey is considered to be out of energy, and stronger attacks are applied with rapid drive progressively. The distance between the target before its acquisition is modeled as:

$$X(t+1) = \begin{cases} Y, & \text{if } F(Y) < F(X(t)) \\ Z, & \text{if } F(Z) < F(X(t)), \end{cases} \qquad (15)$$

for which the $Y$ and $Z$ are obtained by the next two equations:

$$Y = X_{best}(t) - E|JX_{best}(t) - X(t)| \qquad (16)$$
$$Z = Y + S \times LF(D) \qquad (17)$$

## Proposed enhanced Harris hawk optimization algorithm

### New initialization scheme

The applied approach exploits a novel initialization strategy of populations:

$$x_{i,j} = lb_j + \psi \cdot (ub_j - lb_j), \qquad (18)$$

in which the $j$-th component of $i$-th solution is given as $x_{i,j}$, the upper and lower bounds are represented by $ub_j$ and $lb_j$ for the parameter $j$, and a pseudo-random number is drawn between $[0, 1]$ and given as $\psi$.

The quasi-reflection-based learning (QRL) procedure has proven to give results (*Jovanovic et al., 2023b*) where applied with the goal of sarge space enlargement for the case of those generated by the (18). The purpose of the QRL procedure is reflected in the fact that if the observed solution falls in the suboptimal region of the search space, there is a fair chance that its opposite will fall in more promising areas of the search domain, as reported by several authors recently (*Bacanin et al., 2023a*; *Basha et al., 2021*; *Nama, 2022*; *Çelik, 2023*; *Lei et al., 2022*; *Bacanin et al., 2021*; *Xue, 2022*). Hence the $x_j^{qr}$, quasi-reflexive-opposite component for all parameters of a solution $x_j$ is provided as in the following equation:

$$X_j^{qr} = \text{rnd}\left(\frac{lb_j + ub_j}{2}, x_j\right), \qquad (19)$$

while at $\left[\frac{lb_j+ub_j}{2}, x_j\right]$ interval a pseudo-random number is chosen as $rnd$.

### Mechanism for maintaining population diversity

Diversification is observed as a parameter of the convergence/divergence ratio during the search process as in *Cheng & Shi (2011)*.

$L1$ norm (*Cheng & Shi, 2011*) applies two-component diversification for the solutions and the dimensions of the problem. Important information for the search process can be derived from the dimension-wise metric with the $L1$ norm.

The number of total individuals is marked with $m$ and the dimensions number as $n$, the $L1$ norm is given as in Eqs. (20)–(22):

---

**Algorithm 1  QRL pseudo-code initialization scheme.**

1: $P^{init}$ population with $N/2$ solutions created by Eq. (18).

2: $P^{qr}$ population by QRL from $P^{init}$ by Eq. (19).

3: Merge $P^{init}$ and $P^{qr}$ $(P \cup P^{qr})$ resulting in the starting population.

4: Fitness calculation of every solution in $P$

5: $P$ sorted by fitness

---

$$\bar{x} = \frac{1}{m} \sum_{i=1}^{m} x_{ij} \tag{20}$$

$$D_j^p = \frac{1}{N} \sum_{i=1}^{N} |x_{ij} - \bar{x}_j| \tag{21}$$

$$D^p = \frac{1}{n} \sum_{i=1}^{n} D_j^p \tag{22}$$

in which every individual's position mean is represented as $\bar{x}$ vector over all dimensions, the hawk's position vector of diversity as $L1$ norm is shown as $D_j^p$, while the scalar form is shown as $D^p$ for the entire population. Using regular strategies of initialization usually results in higher diversity with weaker convergence towards later iterations. The described metric is used for $L1$ determination of the threshold $D_t$ for the diversity. Firstly, the $D_{t0}$ is calculated by Eq. 23, which is followed by condition $D^P < D_t$ for the satisfactory value of diversity, the worst solutions are replaced with randomly generated solutions $nrs$ with the same strategy for population initialization. The $nrs$ value is another control parameter.

$$D_{t0} = \sum_{j=1}^{n} \frac{(ub_j - lb_j)}{2 \cdot n} \tag{23}$$

The Eq. (1) and Algorithm 1 indicate close generation of solutions towards the bounds of the search space's mean. The value $D_t$ falls of as shown in:

$$D_{t,iter+1} = D_{t,iter} - D_{t,iter} \cdot \frac{iter}{T}, \tag{24}$$

in which the current and subsequent iterations are given as $iter$ and $iter + 1$, and the number of iterations at the maximum is $T$. According to this mechanism, the $D_t$ falls off in no relation to the $D^P$ and still will not trigger the mechanism.

### Inner workings and complexity of proposed method

Taking inspiration from applied mechanisms to the original solution the proposed new algorithm is diversity directed HHO (DDHHO), which is shown in Algorithm 2. It is important to note that the computational complexity of the original algorithm is not lower than that of the novel solution. In modern literature, it is a practice to measure this in FFEs as it is the most resource-demanding technique, hence the complexity of the DDHHO for

**Algorithm 2  Pseudo-code of the basic HHO algorithm implementation.**

**Inputs**: The population size $N$ and maximum number of iterations $T$

**Outputs**: The location of the rabbit and its fitness value

Initialize the random population $X_i(i = 1, 2, \ldots, N)$

Initialize population $X_i$, $(i = 1, 2, 3, \ldots N)$ according to Algorithm 1

Determine values of $D_{t0}$ and $D_t$

**while** (stopping condition is not met) **do**

    Calculate the fitness values of hawks

    Set $X_{rabbit}$ as the location of rabbit (best location)

    **for** (each hawk $(X_i)$) **do**

        Update the initial energy $E_0$ and jump strength $J$

        Update the $E$ using Eq. (6)

        **if** ($|E| \geq 1$) **then**

            Update the location vector using Eq. (4)

        **end if**

        **if** ($|E| < 1$) **then**

            **if** ($r \geq 0.5$ and $|E| \geq 0.5$) **then**

                Update the location vector using Eq. (7)

            **else if** ($r \geq 0.5$ and $|E| < 0.5$) **then**

                Update the location vector using Eq. (10)

            **else if** ($r < 0.5$ and $|E| \geq 0.5$) **then**

                Update the location vector using Eq. (14)

            **else if** ($r < 0.5$ and $|E| < 0.5$) **then**

                Update the location vector using Eq. (15)

            **end if**

        **end if**

    **end for**

    Calculate $D^P$

    **if** ($D^P < D_t$) **then**

        Replace worst $nrs$ with solutions created as in (18)

    **end if**

    Update $D_t$ by expression (24)

**end while**

**Return $X_{rabbit}$**

the worst scenario is *Yang & He (2013)*: $O(DDHHO) = O(N) + O(T \cdot N^2)$. In comparison to other metaheuristics algorithms, the complexity of the DDHHO is similar. For instance, firefly algorithm (*Yang & Slowik, 2020*) is more complex as it evaluates at most $N * N$ solutions in each iteration.

### Hyperparameter optimization using HHO

To optimize the hyperparameters of the Luong attention-based RNN model, we perform the following steps:

**Define the search space:** Identify the hyperparameters to be optimized and specify their respective ranges or discrete sets of possible values. For instance, for the number of hidden layers, we may specify a range of values, *e.g.*, from 1 to 5. Similarly, we define the search space for other hyperparameters such as the number of hidden units per layer, type of RNN cell, attention mechanism, attention scoring function, learning rate, dropout rate, batch size, and sequence length.

**Initialize the population:** Generate an initial population of candidate solutions, where each candidate solution represents a combination of hyperparameter values within the defined search space.

**Evaluate candidate solutions:** For each candidate solution, train the Luong attention-based RNN model using the specified hyperparameter values, and evaluate the performance on a validation set using one or more performance metrics (*e.g.*, MAE, RMSE, and MAPE). This step may require cross-validation or other validation techniques to obtain reliable performance estimates.

**Apply optimization algorithm:** Utilize the chosen metaheuristic optimization algorithm for search space exploration and find the best combination of hyperparameter values that minimizes the chosen performance metric(s). In each iteration, the algorithm updates the candidate solutions based on the optimization strategy specific to the chosen algorithm, and the performance of the updated solutions is re-evaluated on the validation set.

**Termination condition:** The optimization process is ongoing until a predefined termination condition is met, such as a maximum iteration number, a minimum performance improvement threshold, or a predefined computational budget.

**Select the optimal solution:** Once the termination condition is reached, select the candidate solution with the best performance on the validation set as the optimal combination of hyperparameter values for the Luong attention-based RNN model.

**Final model training and evaluation:** Train the Luong attention-based RNN model using the optimal hyperparameter values on the entire training set, and evaluate its performance on the test set to obtain an unbiased estimate of the model's forecasting accuracy.

## DATASET DESCRIPTION AND EXPERIMENTS

This section aims to provide an overview of the datasets utilized in the experiments and the experimental setup established for all methods employed in the comparative analysis.

### Utilized datasets

#### Spain solar energy dataset

The first dataset, concerning photovoltaic power generation in Spain, is constructed from real-world originating from two different sources. The ENTSO-E portal (https://transparency.entsoe.eu/) provides hourly energy demand and generation considering the

renewable energy in Spain, while the weather data is provided by OpenWeather API (https://openweathermap.org/guide) for the location of Valencia, Spain.

Considering the large amount of data available, a smaller dataset segment was utilized during experimentation. The datasets cover hourly data from 1.8.2018. to 31.12.2018. and covered a total of 3,670 data points. The hourly metrics that were the most relevant are included for multivariate forecasting as well as the data and support metrics of generated photovoltaic power. The dataset was then further separated and with 70% of the data used for training, 10% for validation, and the remaining 20% for testing. The included features include generated photovoltaic power, as well as humidity, rainfall, cloud cover, and ambient temperature. With the generated photovoltaic power feature being the prediction target.

### China wind farm dataset

The Global Energy Forecasting Competition 2012 (GEFCom2012) is a competition that aimed to promote the development of state-of-the-art forecasting models for various aspects of the energy industry. The dataset related to wind farms in China used in a competition (https://www.kaggle.com/competitions/global-energy-forecasting-competition-2012-load-forecasting/data). Seven wind farms from mainland China were selected and anonymized for this dataset. Power generation data has been normalized as well due to anonymity concerns.

Relevant wind data is collected every 12 h while the dataset includes forecasts in intervals of 24 h. The direction and speed of the wind and meridional wind components are provided as well. The dataset consists of hourly measurements of wind power generation from seven wind farms located in China, spanning from January 1, 2011, to September 30, 2012. Each wind farm has different installed capacities, which makes the forecasting task more challenging. For experimentation, hourly resolution data has been split into predictions of 12 h and then further combined with normalized real-world data of power generation for each farm by the hour. Due to the last year of data not being available, the dataset consists of four years of data. The included features are Wind speed, wind direction, and zonal and meridional wind components for each wind farm while the target feature is the amount of generated power.

The first 70% of the available data points were utilized for training, while the later 10% and 20% were used for validation and testing.

### Data preprocessing

Before using the dataset for renewable power generation forecasting, some preprocessing steps may be necessary:

1. **Missing data imputation:** The dataset may contain values that are missing, which are required to be imputed before using the data for model training and evaluation. Various imputation techniques can be employed, such as linear interpolation or more advanced methods based on machine learning models.

2. **Data splitting:** The division of the dataset into training, validation, and testing subsets. The training and validation sets can be used for model development and hyperparameter

tuning, while the testing set can be used for final performance evaluation of the model's forecasting.

3. **Feature engineering:** Extract additional features from the dataset that may be relevant for the forecasting task, such as lagged values of wind power, moving averages, or other temporal features that can help in pattern and dependendcy caputring in the data.

4. **Normalization/standardization:** Scale the input features and target variable to ensure that they are on a similar scale, which is able of improving the performance and stability.

Once the dataset is preprocessed, it can be used to train and evaluate various forecasting models, such as the Luong attention-based RNN model discussed earlier. By incorporating techniques like time-series decomposition, attention mechanisms, and hyperparameter optimization, the forecasting models can be tailored to the specific characteristics and challenges of the wind power generation data, ultimately improving the accuracy and reliability of the forecasts.

## Experimental setup

The following setup regards all four test cases that have been executed. Two stages are differentiated during experimentation. During the first, the data is decomposed for both test cases. Afterward, the signal components and residual signals are provided to the RNN for forecasting. VMD was employed for feature engineering, and min-max scaling was utilized as scaling option. Every tested model was provided in the same manner with historic data of six input points per model for three steps ahead predictions.

The data was split in the same manner for all four test cases, with the training set amounting to 70%, the validation set of 10%, and the testing set of 20%. The split of each the solar dataset target features is visualized with Fig. 1 to illustrate the time intervals that were employed in each of the three mentioned subsets. Similarly, the wind dataset is shown in Fig. 2.

The challenge of parameter optimization for the prediction models was tested on the following contemporary metaheuristics: GA (*Mirjalili & Mirjalili, 2019*), PSO (*Kennedy & Eberhart, 1995*), ABC (*Karaboga, 2010*), FA (*Yang & Slowik, 2020*), COLSHADE (*Gurrola-Ramos, Hernàndez-Aguirre & Dalmau-Cedeño, 2020*), and self-adaptive step size algorithm (*Tang & Gibali, 2020*). Additionally, to the mentioned metaheuristics the original HHO and the DDHHO were evaluated. Each algorithm was executed with eight solutions in the population and five iterations.

The parameters for the VMD were empirically established and the parameter $K = 3$, while the *alpha* parameter represents the length of the used dataframe. To ensure the objectivity of model evaluation 30 independent runs were performed due to the stochastic nature of the optimization algorithms. The selected parameters for optimization of the RNN are given in the following text due to their impact on the performance of the model. The ranges of the parameters alongside their descriptions are given: [50, 100] number of neurons, [0.0001, 0.01] learning rate, [100, 300] training epochs, [0.05, 0.1] dropout rate, and [1, 3] for the total layer number of a network.

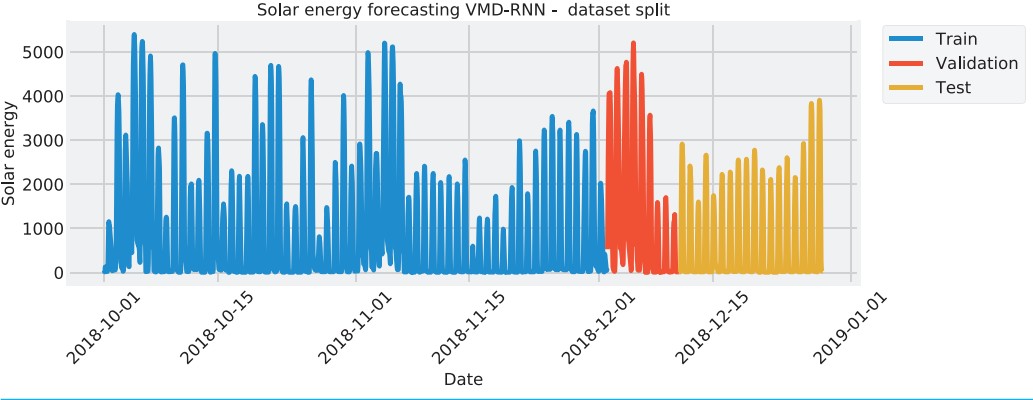

**Figure 1 Solar energy generation target feature split.**

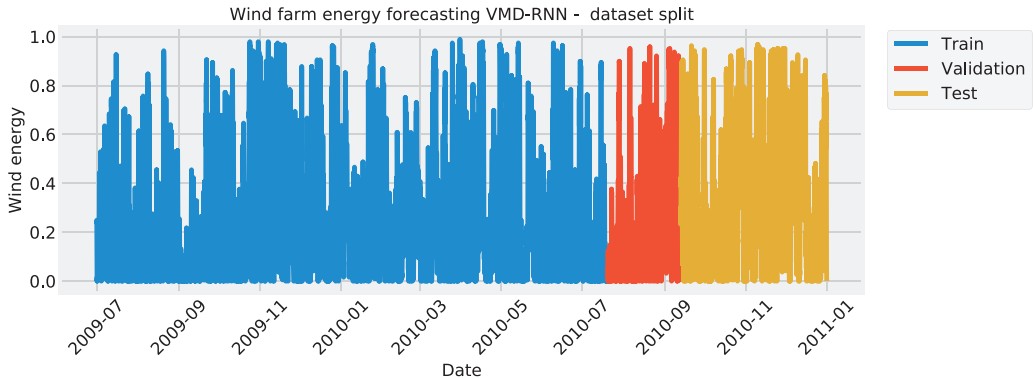

**Figure 2 Wind energy generation target feature Split.**

Lastly, an early stopping mechanism is incorporated for overfitting prevention with the threshold empirically determined as $\frac{epochs}{3}$. The purpose of such a mechanism is to terminate the model early if no improvements are observed for $\frac{epochs}{3}$. It should be noted that computational resource waste is reduced as an effect of this approach.

This study employs five performance metrics commonly used to evaluate the accuracy and effectiveness of the proposed attention-based recurrent neural network (A-RNN) model for renewable power generation forecasting. These performance metrics are mean absolute error (MAE), root mean squared error (RMSE), mean absolute error (MAE), Coefficient of determination ($R^2$) and the index of alignment (IA).

MAE is the average of the absolute differences between the predicted values and the actual values. It measures the magnitude of errors in the forecasts without considering their direction. The MAE is defined as:

$$MAE = \frac{1}{N} \sum_{i=1}^{N} |y_i - \hat{y}_i| \tag{25}$$

for which the $N$ represents data points total, $y_i$ the actual value, and $\hat{y}_i$ the predicted value.

RMSE is the square root of the average of the squared differences between the predicted values and the actual values. It provides a measure of the overall model's performance by penalizing larger errors more than smaller errors. The RMSE is defined as:

$$RMSE = \sqrt{\frac{1}{N} \sum_{i=1}^{N} (y_i - \hat{y}_i)^2}. \tag{26}$$

MAE is the average of the absolute differences between the predicted values and the actual values. It can be useful for comparing the performance of different models across various scales. The MAE is defined as:

$$MAE = \frac{1}{n} \sum_{i=1}^{n} y_i - \hat{y}_i \tag{27}$$

where the $||$ denotes the absolute value.

$R^2$ indicates the proportion of the variance in the dependent variable that can be explained by the independent variables in the model. It ranges from 0 to 1, with higher values indicating a better fit between the model and the data. $R^2$ is defined as:

$$R^2 = 1 - \frac{\sum_i (y_i - \hat{y}_i)^2}{\sum_i (y_i - \bar{y})^2} \tag{28}$$

where the $\bar{y}$ refers to the mean of the actual values.

IA measures the extent to which the model's predicted outcomes align with the true outcomes or the intended goals. A higher Alignment Index indicates a stronger alignment, suggesting that the model is performing well. AI is defined as:

$$IA = 1 - \frac{\sum_{i=1}^{n} (y_i - \hat{y}_i)^2}{\sum_{i=1}^{n} (|y_p - \bar{y}| + |y_i = \bar{y}|)^2}. \tag{29}$$

These performance metrics, MAE, RMSE, and MAPE, are used to evaluate the accuracy and effectiveness of the proposed A-RNN model in comparison to the regular RNN model for renewable power generation forecasting. A lower value for each metric indicates better forecasting performance.

A flowchart of the utilized experimental framework is provided in Fig. 3.

## RESULTS AND COMPARISON

This section exhibits obtained experimental findings in terms of captured performance metrics. The best metrics in all tables were marked with bold style to more clearly visualize the best performing methods.

### Spain solar energy forecasting

In Table 1 the objective function outcomes for the best, worst, mean, and median executions, alongside the standard deviance with variance are shown for 30 independent runs of each metaheuristic.

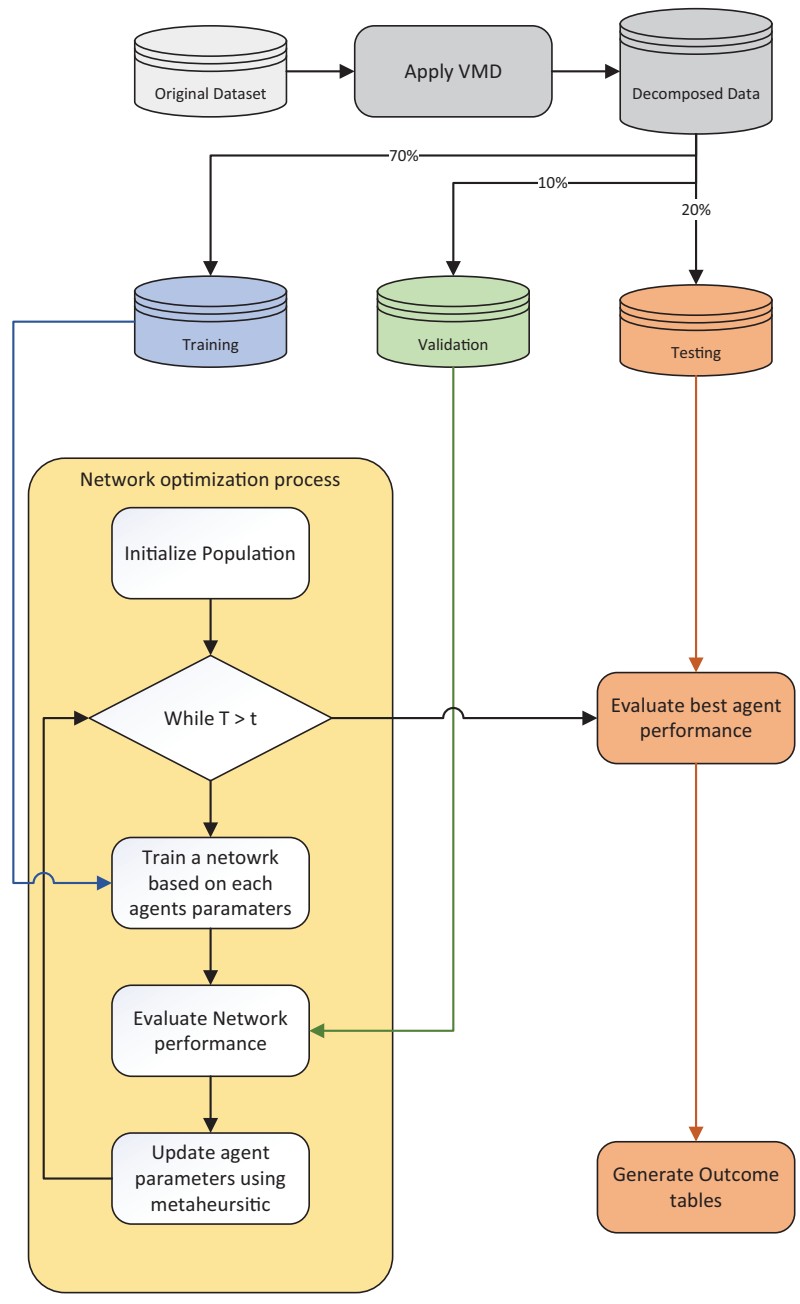

**Figure 3 Experimental framework flowchart.**

As Table 1 suggests, the introduces algorithms attained the best results when optimizing a RNN in the best run. However, admirable stability was demonstrated by the PSO. Furthermore, when considering the worst case execution the ABC attained the best results as well as in the mean and median runs. This is to be expected as per the NFL (*Wolpert & Macready, 1997*) no single approach works equally well in all execution cases.

**Table 1  VMD-RNN solar energy forecasting objective function overall outcomes.**

| Method | Best | Worst | Mean | Median | Std | Var |
|---|---|---|---|---|---|---|
| VMD-RNN-DDHHO | **0.006284** | 0.007320 | 0.006855 | 0.006931 | 0.000389 | 1.513667E-7 |
| VMD-RNN-HHO | 0.006990 | 0.007890 | 0.007366 | 0.007282 | 0.000344 | 1.183526E-7 |
| VMD-RNN-GA | 0.006664 | 0.007559 | 0.007061 | 0.007228 | 0.000341 | 1.163809E-7 |
| VMD-RNN-PSO | 0.007186 | 0.007458 | 0.007345 | 0.007425 | **0.000115** | **1.320113E-8** |
| VMD-RNN-ABC | 0.006499 | **0.007231** | **0.006830** | **0.006801** | 0.000251 | 6.319240E-8 |
| VMD-RNN-FA | 0.007005 | 0.007542 | 0.007184 | 0.007014 | 0.000229 | 5.253891E-8 |
| VMD-RNN-COLSHADE | 0.007159 | 0.008009 | 0.007478 | 0.007182 | 0.000357 | 1.273813E-7 |
| VMD-RNN-SASS | 0.007057 | 0.007405 | 0.007264 | 0.007240 | 0.000135 | 1.829039E-8 |

**Table 2  The VMD-RNN solar energy metrics per each step.**

| Step | Metric | VMD-RNN-DDHHO | VMD-RNN-HHO | VMD-RNN-GA | VMD-RNN-PSO | VMD-RNN-ABC | VMD-RNN-FA | VMD-RNN-COLSHADE | VMD-RNN-SASS |
|---|---|---|---|---|---|---|---|---|---|
| One step | $R^2$ | 0.601739 | 0.549365 | **0.627364** | 0.528460 | 0.58500 | 0.544636 | 0.543719 | 0.559259 |
| | MAE | **384.294171** | 432.200603 | 396.006180 | 427.516283 | 404.377133 | 418.018708 | 411.089031 | 412.655917 |
| | MSE | 400,081.633100 | 452,694.787317 | **374,338.747453** | 473,694.873874 | 416,895.063424 | 457,445.578253 | 458,366.263037 | 442,755.336455 |
| | RMSE | 632.520065 | 672.825971 | 611.832287 | **688.254948** | 645.674115 | 676.347232 | 677.027520 | 665.398630 |
| | IA | 0.886044 | 0.870430 | **0.896802** | 0.870911 | 0.877714 | 0.875709 | 0.875988 | 0.877386 |
| Two step | $R^2$ | **0.8896686** | 0.878472 | 0.844966 | 0.868775 | 0.876350 | 0.885817 | 0.873014 | 0.8760918 |
| | MAE | **195.801662** | 227.673953 | 246.869567 | 233.834781 | 227.774440 | 204.845965 | 216.919114 | 219.607867 |
| | MSE | **110,835.615218** | 122,082.984352 | 155,742.443523 | 131,825.249471 | 124,214.713878 | 114,704.662015 | 127,566.656326 | 124,474.546886 |
| | RMSE | **332.919833** | 349.403755 | 394.642172 | 363.077470 | 352.441079 | 338.680767 | 357.164747 | 352.809505 |
| | IA | **0.970558** | 0.966796 | 0.960179 | 0.966048 | 0.965562 | 0.969940 | 0.968305 | 0.966947 |
| Three step | $R^2$ | 0.962557 | 0.964848 | 0.948636 | 0.978350 | 0.973942 | **0.960881** | 0.961240 | 0.951496 |
| | MAE | 122.562368 | 137.209296 | 165.046855 | **105.082911** | 112.980142 | 141.060131 | 124.093137 | 141.036372 |
| | MSE | 37,613.696545 | 35,313.037867 | 51,598.255163 | **21,749.216531** | 26,177.198226 | 39,297.213129 | 38,936.684159 | 48,725.218704 |
| | RMSE | 193.942508 | 187.917636 | 227.152493 | **147.4761560** | 161.793690 | 198.235247 | 197.323805 | 220.737896 |
| | IA | 0.9901459 | 0.990594 | 0.986690 | **0.994450** | 0.992991 | 0.989871 | 0.990657 | 0.987153 |
| Overall | $R^2$ | 0.817988 | 0.797562 | 0.806989 | **0.791861** | 0.811765 | 0.797111 | 0.792658 | 0.795616 |
| | MAE | **234.219400** | 265.694617 | 269.307534 | 255.477992 | 248.377238 | 254.641602 | 250.700427 | 257.766719 |
| | MSE | **182,843.648288** | 203,363.603179 | 193,893.148713 | 209,089.779959 | 189,095.658509 | 203,815.817799 | 208,289.867841 | 205,318.367348 |
| | RMSE | **427.602208** | 450.958538 | 440.332998 | 457.263360 | 434.851306 | 451.459652 | 456.387848 | 453.120698 |
| | IA | **0.948916** | 0.942607 | 0.947890 | 0.943803 | 0.945423 | 0.945173 | 0.944983 | 0.943829 |

**Note:**
The best results are shown in bold.

Further detailed metrics for the best run, for each forecasting step and every tested metaheuristic are demonstrated in Table 2.

As it can be observed from Table 2 the introduced method attained the best overall results in all cases except the $R^2$ metric, where the PSO attained better results. As the guiding objective function during the optimization process was MSE this is to be expected. Additionally the introduced method also attained the best results when making forecasts two steps ahead, as well MAE for one step ahead. The best results for $R^2$, MSE and IA where attained by the GA, while the best RMSE results where attained by the PSO.

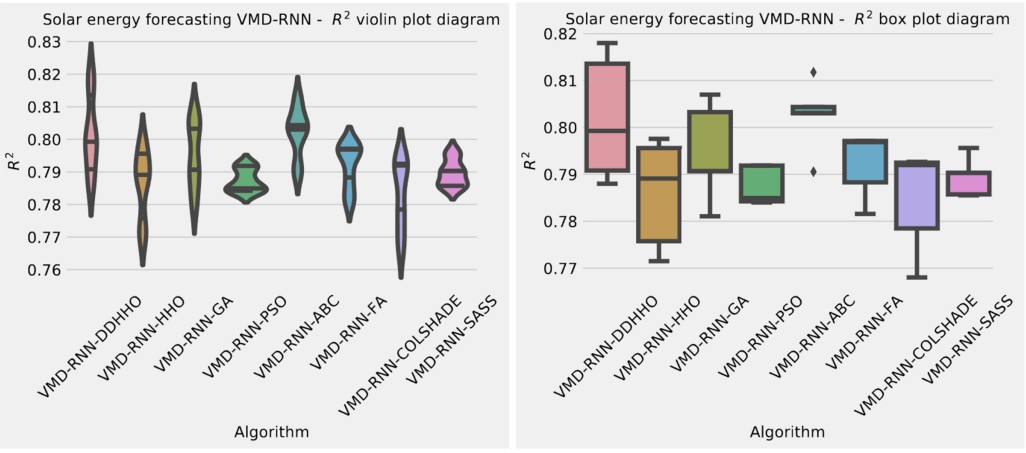

**Figure 4 Solar dataset objective function and $R^2$ distribution plots for each metaheurstic without attention layer.**

Nevertheless when making forecasts three steps ahead the PSO attained the best results across all metrics except $R^2$ where the FA attained the best outcomes.

To help demonstrated the improvements made by the introduced method visualizations are provided for the distribution of both MSE and $R^2$ are shown in Fig. 4 followed by convergence plots for both functions in Fig. 5 and swarm and KDE plots in Fig. 6.

Finally, the parameters selected by each metaheuristic for their respective best models are shown in Table 3.

Similarly to the previous experiment, in Table 4 the objective function outcomes for the best, worst, mean, and median executions, alongside the standard deviance with variance are shown for 30 independent runs of each metaheuristic.

Interestingly, when optimizing the RNN-ATT models, the introduced metaheuristic demonstrated better performance overall most metrics. However, the ABC and SASS algorithms demonstrated a slightly higher degree of stability despite attaining less impressive results.

Further detailed metrics for the best run, for each forecasting step and every tested metaheuristic are demonstrated in Table 5.

As it can be observed in Table 5 the introduces method attained the best overall results for MSE and MAE, while the HHO attained the best IA results, the ABC attained the best $R^2$ outcomes overall, while SASS attained the best outcomes for MAE. The introduced approach demonstrated the best performance when making predictions one step ahead, while two step ahead forecasts are done best by the PSO. No single approach performed the best for three steps ahead, while different metaheuristics attaining first place in different metrics further enforcing the NFL (*Wolpert & Macready, 1997*) theorem.

Visualizations of objective function and $R^2$ distributions are shown in Fig. 7 followed by their respective convergence graphs in Fig. 8. The KDE and swarm plots are also provided in Fig. 9.

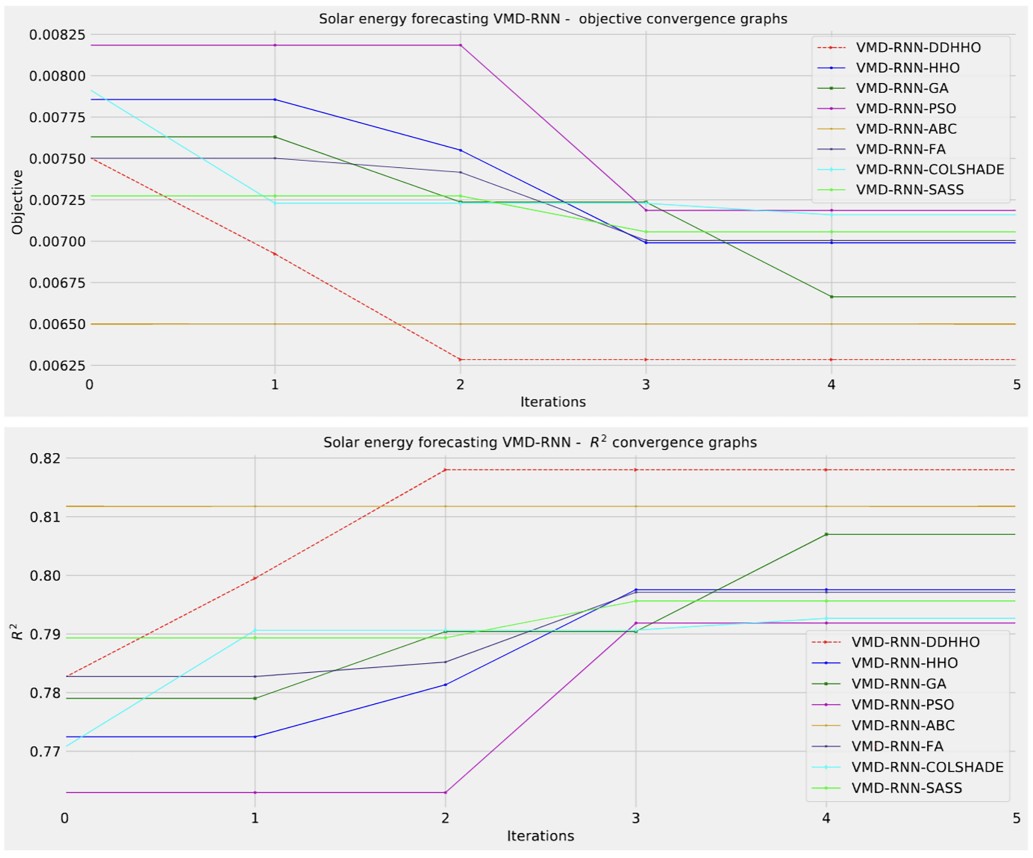

**Figure 5 Solar dataset objective function and $R^2$ convergence plots for each metaheuristic without attention layer.**

The parameters selected by each competing metaheuristic for their respective best-performing models are shown in Table 6.

In Table 7 the objective function outcomes for the best, worst, mean, and median executions, alongside the standard deviance with variance are shown for 30 independent runs of each metaheuristic forecasting wind power generation.

## China wind farm forecasting

The introduced metaheuristic attained the best outcomes in the best, mean and median executions, with the ABC attained the best outcomes in the worst case executions. Furthermore, the highest stability was demonstrated by SASS.

Further detailed metrics for the best run, for each forecasting step and every tested metaheuristic are demonstrated in Table 8.

As demonstrated in Table 8, the introduced metaheursitic outperformed all competing metaheuristic in overall outcomes. THe introduces metaheuristic demonstrated the best results for one step ahead forecasts¿ However, the PSO attained the best results for two steps ahead forecasts, and COLSHADE attained the best outcomes for three steps ahead. These results further reinforce that no single approach is equally suited to all use-cases as

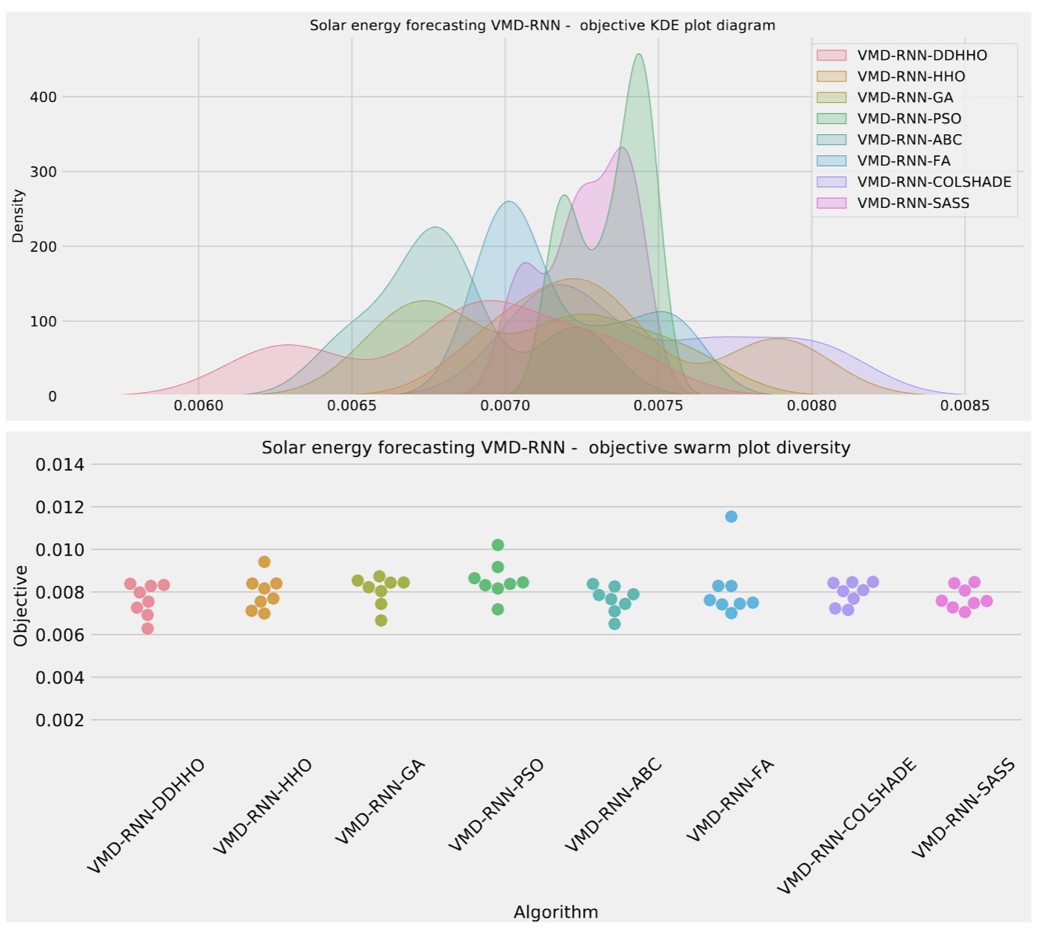

**Figure 6 Solar dataset objective swarm and KDE plots for each metaheuristic without attention layer.**

**Table 3 Parameters for best performing solar prediction RNN model optimized by each metaheuristic.**

| Method | Learning rate | Dropout | Epochs | Layers | L1 Neurons | L2 Neurons | L3 Neurons |
|---|---|---|---|---|---|---|---|
| VMD-RNN-DDHHO | 0.007050 | 0.050000 | 232 | 3 | 50 | 100 | 100 |
| VMD-RNN-HHO | 0.007349 | 0.076853 | 206 | 3 | 64 | 50 | 100 |
| VMD-RNN-GA | 0.009097 | 0.091104 | 114 | 2 | 89 | 52 | / |
| VMD-RNN-PSO | 0.009329 | 0.069591 | 223 | 2 | 69 | 89 | / |
| VMD-RNN-ABC | 0.010000 | 0.100000 | 181 | 3 | 92 | 64 | 79 |
| VMD-RNN-FA | 0.010000 | 0.088052 | 238 | 2 | 50 | 50 | / |
| VMD-RNN-COLSHADE | 0.008718 | 0.063527 | 288 | 3 | 85 | 100 | 100 |
| VMD-RNN-SASS | 0.006645 | 0.096538 | 300 | 3 | 100 | 86 | 54 |

per the NFL (*Wolpert & Macready, 1997*). Visualizations of the distribution and convergence rates of the mse and $R^2$ functions are shown in Figs. 10 and 11. Additionally, KDE and swarm diversity plots are provided in Fig. 12.

**Table 4  VMD-RNN-ATT solar energy forecasting objective function overall outcomes.**

| Method | Best | Worst | Mean | Median | Std | Var |
|---|---|---|---|---|---|---|
| VMD-RNN-ATT-DDHHO | **0.006517** | **0.007211** | **0.006923** | **0.006944** | 0.000250 | 6.265266E-8 |
| VMD-RNN-ATT-HHO | 0.007036 | 0.008443 | 0.007447 | 0.007111 | 0.000613 | 3.759833E-7 |
| VMD-RNN-ATT-GA | 0.006705 | 0.008075 | 0.007389 | 0.007209 | 0.000499 | 2.490886E-7 |
| VMD-RNN-ATT-PSO | 0.006711 | 0.007571 | 0.007233 | 0.007303 | 0.000297 | 8.818285E-8 |
| VMD-RNN-ATT-ABC | 0.007452 | 0.007531 | 0.007480 | 0.007470 | 0.000032 | **1.025433E-9** |
| VMD-RNN-ATT-FA | 0.007222 | 0.008049 | 0.007641 | 0.007647 | 0.000292 | 8.550797E-8 |
| VMD-RNN-ATT-COLSHADE | 0.006915 | 0.007912 | 0.007455 | 0.007476 | 0.000363 | 1.318140E-7 |
| VMD-RNN-ATT-SASS | 0.007238 | 0.007720 | 0.007472 | 0.007432 | **0.000164** | 2.673677E-8 |

Note:
The best results are shown in bold.

**Table 5  The VMD-RNN-ATT solar energy metrics per each step.**

| Step | Metric | VMD-RNN-ATT-DDHHO | VMD-RNN-ATT-HHO | VMD-RNN-ATT-GA | VMD-RNN-ATT-PSO | VMD-RNN-ATT-ABC | VMD-RNN-ATT-FA | VMD-RNN-ATT-COLSHADE | VMD-RNN-ATT-SASS |
|---|---|---|---|---|---|---|---|---|---|
| 1 | $R^2$ | **0.715471** | 0.584499 | 0.598188 | 0.574065 | 0.603103 | 0.548291 | 0.616813 | 0.547094 |
|  | MAE | **376.979586** | 442.064510 | 462.047919 | 435.538303 | 474.267738 | 435.524720 | 423.718303 | 416.220384 |
|  | MSE | **285,829.818133** | 417,399.667275 | 403,648.569532 | 427,881.634339 | 398,711.291244 | 453,773.352978 | 384,938.817726 | 454,976.265366 |
|  | RMSE | **534.630544** | 646.064755 | 635.333432 | 654.126620 | 631.435896 | 673.627013 | 620.434378 | 674.519285 |
|  | IA | **0.9146240** | 0.889628 | 0.881474 | 0.871310 | 0.891814 | 0.873488 | 0.887386 | 0.861529 |
| 2 | $R^2$ | 0.829019 | 0.876223 | 0.874955 | **0.888033** | 0.837797 | 0.868852 | 0.874406 | 0.861896 |
|  | MAE | 252.954113 | 243.425326 | 260.158326 | **218.732420** | 290.688281 | 236.760030 | 252.883363 | 233.639125 |
|  | MSE | 171,762.088320 | 124,342.580437 | 125,616.779871 | **112,478.817327** | 162,944.638909 | 131,747.397307 | 126,168.484562 | 138,735.683810 |
|  | RMSE | 414.441900 | 352.622433 | 354.424576 | **335.378618** | 403.6640174 | 362.970243 | 355.202033 | 372.472393 |
|  | IA | 0.951127 | **0.967796** | 0.965910 | 0.967226 | 0.958823 | 0.966348 | 0.966094 | 0.961092 |
| 3 | $R^2$ | **0.889236** | 0.927962 | 0.9442501 | 0.954781 | 0.911610 | 0.955364 | 0.907969 | 0.962090 |
|  | MAE | 244.240630 | 219.831502 | 179.063882 | **144.828299** | 232.407156 | 154.496558 | 244.166959 | 131.982225 |
|  | MSE | 111,269.990578 | 72,366.697870 | 56,004.659587 | 45,425.756743 | 88,793.700643 | **44,840.040944** | 92,451.964057 | 38,082.907643 |
|  | RMSE | 333.571567 | 269.010590 | 236.653036 | 213.133190 | 297.982719 | 211.754672 | 304.059146 | **195.14842** |
|  | IA | 0.968308 | 0.980827 | 0.985080 | 0.987410 | 0.976862 | 0.988566 | 0.974996 | **0.989529** |
| Overall | $R^2$ | 0.811242 | 0.796228 | 0.805798 | 0.805626 | **0.784170** | 0.790836 | 0.799729 | 0.790360 |
|  | MAE | 291.391443 | 301.77378 | 300.423376 | 266.366341 | 332.454391 | 275.593769 | 306.922875 | **260.613911** |
|  | MSE | **189,620.632344** | 204,702.981861 | 195,090.002997 | 195,262.069470 | 21,6816.543599 | 210,120.263743 | 20,1186.422115 | 210,598.285607 |
|  | RMSE | **435.454512** | 452.441136 | 441.689940 | 441.884679 | 465.635634 | 458.388769 | 448.538094 | 458.909888 |
|  | IA | 0.944686 | **0.946083** | 0.944154 | 0.941982 | 0.942500 | 0.942801 | 0.942826 | 0.937383 |

Note:
The best results are shown in bold.

The network hyperparameters selected by each metaheuristic for the respective best performing models are shown in Table 9.

Similarly to the previous experiment, in Table 10 the objective function outcomes for the best, worst, mean, and median executions, alongside the standard deviance with variance are shown for 30 independent runs of each metaheuristic.

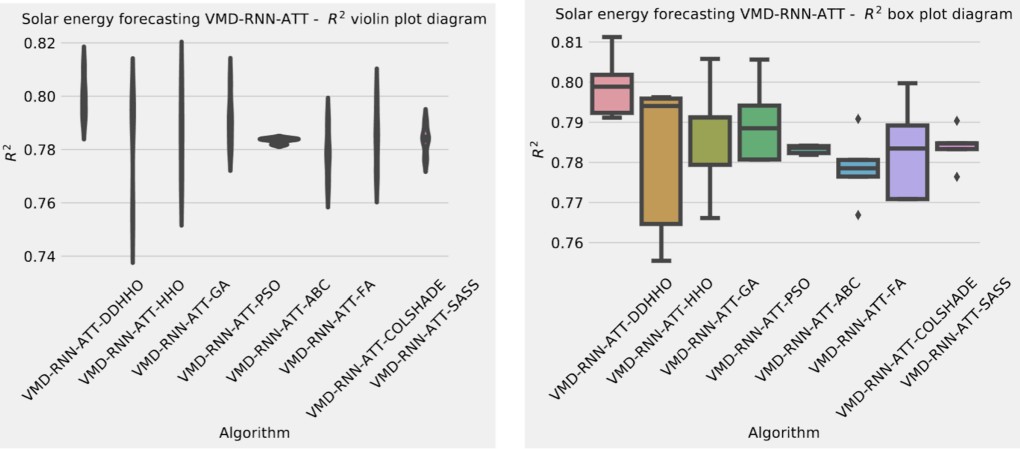

**Figure 7 Solar dataset objective function and $R^2$ distribution plots for each metaheurstic with attention layer.**

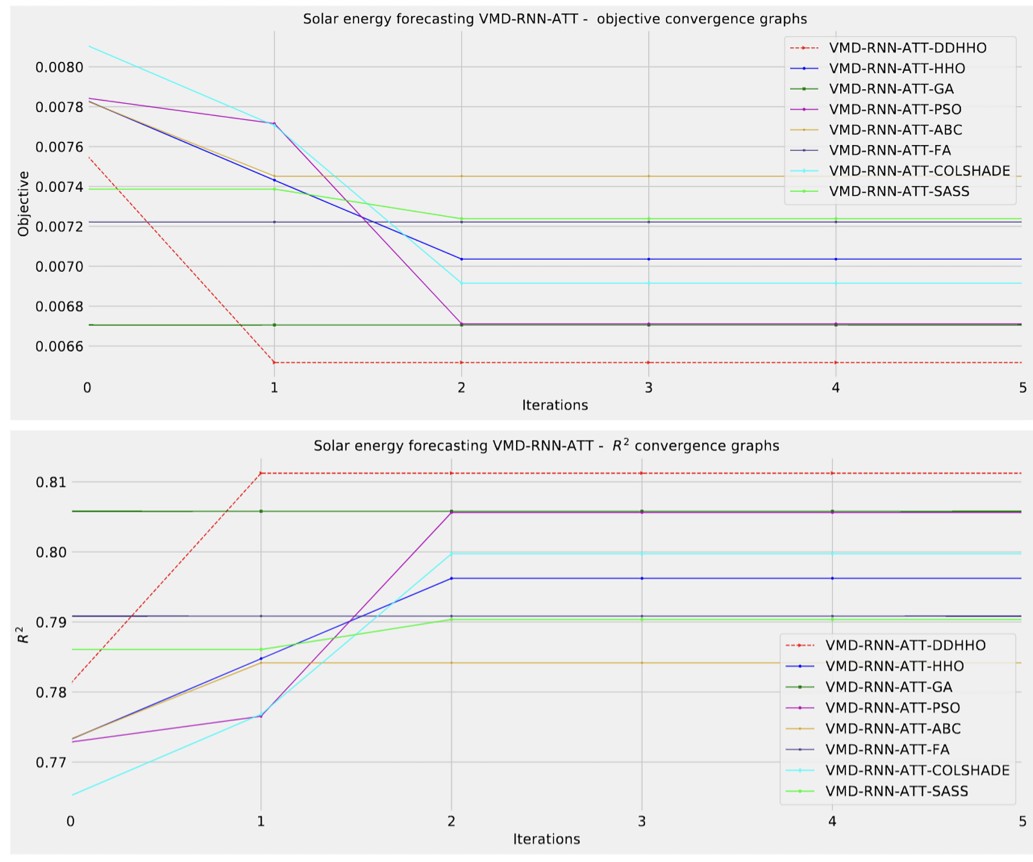

**Figure 8 Solar dataset objective function and $R^2$ convergence plots for each metaheuristic with attention layer.**

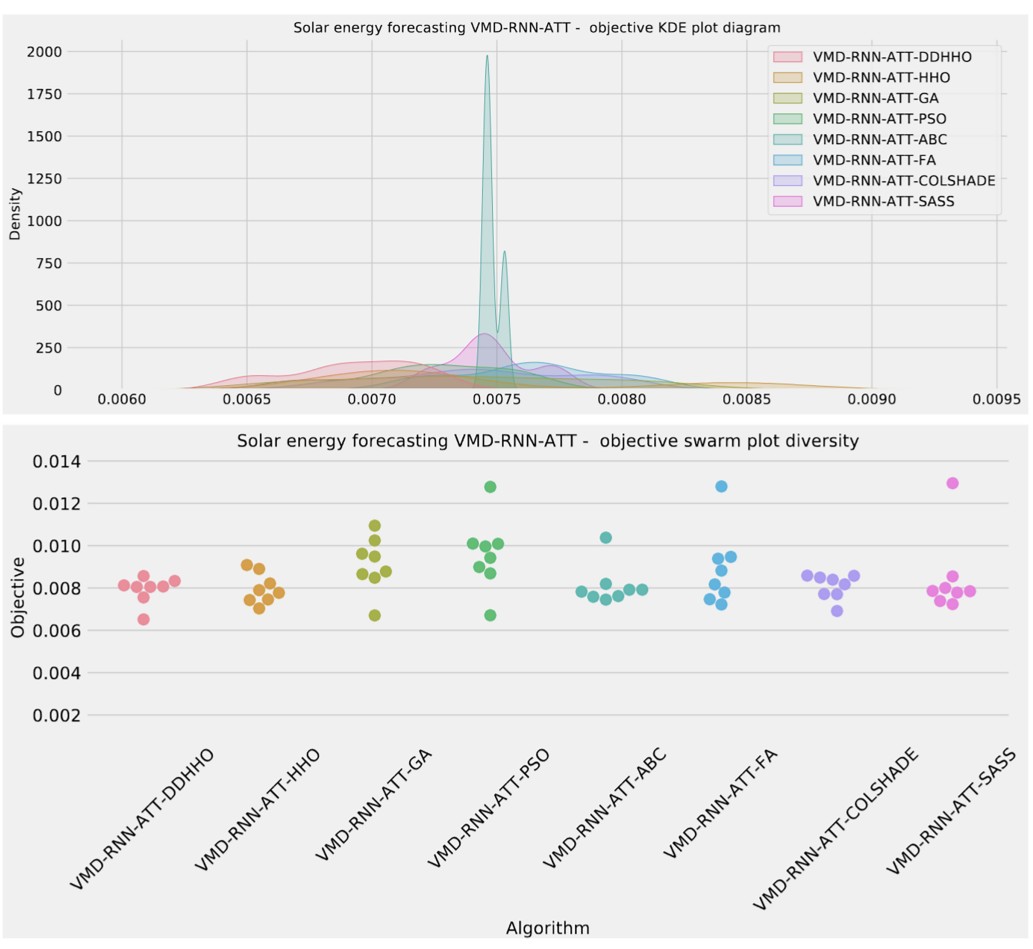

**Figure 9 Solar dataset objective swarm and KDE plots for each metaheuristic with attention layer.**

**Table 6 Parameters for best performing solar prediction RNN-ATT model optimized by each metaheuristic.**

| Method | Learning Rate | Dropout | Epochs | Layers | L1 Neurons | L2 Neurons | L3 Neurons | ATT Neurons |
|---|---|---|---|---|---|---|---|---|
| VMD-RNN-ATT-DDHHO | 0.010000 | 0.100000 | 100 | 3 | 100 | 100 | 50 | 50 |
| VMD-RNN-ATT-HHO | 0.009323 | 0.100000 | 100 | 1 | 98 | / | / | 50 |
| VMD-RNN-ATT-GA | 0.009990 | 0.080219 | 148 | 2 | 71 | 69 | / | 82 |
| VMD-RNN-ATT-PSO | 0.008559 | 0.097184 | 166 | 3 | 89 | 51 | 99 | 96 |
| VMD-RNN-ATT-ABC | 0.010000 | 0.067651 | 101 | 1 | 50 | / | / | 50 |
| VMD-RNN-ATT-FA | 0.006927 | 0.052260 | 216 | 2 | 90 | 87 | / | 97 |
| VMD-RNN-ATT-COLSHADE | 0.004221 | 0.050000 | 120 | 1 | 50 | / | / | 71 |
| VMD-RNN-ATT-SASS | 0.009982 | 0.099805 | 188 | 3 | 100 | 50 | 50 | 50 |

As it can be observed in Table 10 the introduced metaheuristic attained the best outcomes in all except the medial case, where the ABC algorithms attained the best results. Further detailed metrics for the best run, for each forecasting step and every tested metaheuristic are demonstrated in Table 11.

**Table 7  VMD-RNN wind energy forecasting objective function overall outcomes.**

| Method | Best | Worst | Mean | Median | Std | Var |
|---|---|---|---|---|---|---|
| VMD-RNN-DDHHO | **0.010465** | 0.011162 | **0.010747** | **0.010764** | 0.000244 | 5.930160E-8 |
| VMD-RNN-HHO | 0.011407 | 0.011707 | 0.011538 | 0.011517 | 0.000125 | 1.559006E-8 |
| VMD-RNN-GA | 0.011028 | 0.011461 | 0.011240 | 0.011256 | 0.000168 | 2.812603E-8 |
| VMD-RNN-PSO | 0.011000 | 0.011507 | 0.011258 | 0.011294 | 0.000186 | 3.459674E-8 |
| VMD-RNN-ABC | 0.010729 | **0.010977** | 0.010847 | 0.010834 | 0.000108 | 1.176703E-8 |
| VMD-RNN-FA | 0.010519 | 0.011483 | 0.011102 | 0.011134 | 0.000381 | 1.448697E-7 |
| VMD-RNN-COLSHADE | 0.010823 | 0.011382 | 0.011214 | 0.011341 | 0.000241 | 5.784354E-8 |
| VMD-RNN-SASS | 0.011042 | 0.011300 | 0.011231 | 0.011298 | **0.000100** | **9.963395E-9** |

**Note:**
The best results are shown in bold.

**Table 8  The VMD-RNN wind energy metrics per each step.**

| Step | Metric | VMD-RNN-DDHHO | VMD-RNN-HHO | VMD-RNN-GA | VMD-RNN-PSO | VMD-RNN-ABC | VMD-RNN-FA | VMD-RNN-COLSHADE | VMD-RNN-SASS |
|---|---|---|---|---|---|---|---|---|---|
| One step | $R^2$ | **0.875214** | 0.855404 | 0.856190 | 0.849434 | 0.861770 | 0.872224 | 0.857508 | 0.861647 |
| | MAE | **0.077761** | 0.084168 | 0.083139 | 0.084909 | 0.081714 | 0.078881 | 0.083685 | 0.081844 |
| | MSE | **0.012012** | 0.013919 | 0.013843 | 0.014494 | 0.013306 | 0.012300 | 0.013716 | 0.013318 |
| | RMSE | **0.109599** | 0.117979 | 0.117658 | 0.120390 | 0.115352 | 0.110905 | 0.117117 | 0.115404 |
| | IA | **0.967674** | 0.960717 | 0.961990 | 0.958739 | 0.962434 | 0.966699 | 0.962278 | 0.962725 |
| Two step | $R^2$ | 0.897775 | 0.892783 | 0.900496 | **0.903051** | 0.900259 | 0.902827 | 0.899419 | 0.899132 |
| | MAE | 0.070751 | 0.074085 | 0.070576 | **0.070070** | 0.070933 | 0.070237 | 0.071078 | 0.071742 |
| | MSE | 0.009840 | 0.010321 | 0.009578 | **0.009332** | 0.009601 | 0.009354 | 0.009682 | 0.009710 |
| | RMSE | 0.099198 | 0.101592 | 0.097869 | **0.096605** | 0.097986 | 0.096716 | 0.098397 | 0.098538 |
| | IA | 0.973272 | 0.971041 | 0.973894 | 0.974067 | 0.973158 | **0.974287** | 0.973057 | 0.973069 |
| Three step | $R^2$ | 0.908009 | 0.904098 | 0.907150 | 0.9121979 | 0.910908 | 0.904295 | **0.913157** | 0.902638 |
| | MAE | 0.067910 | 0.071199 | 0.069404 | 0.0681129 | 0.068257 | 0.070842 | **0.066382** | 0.072017 |
| | MSE | 0.008855 | 0.009232 | 0.008938 | 0.0084520 | 0.008576 | 0.009213 | **0.008360** | 0.009372 |
| | RMSE | 0.094102 | 0.096081 | 0.094540 | 0.0919348 | 0.092607 | 0.095982 | **0.091431** | 0.096810 |
| | IA | 0.975517 | 0.974068 | 0.975414 | 0.9765410 | 0.976470 | 0.974705 | **0.976785** | 0.973296 |
| Overall | $R^2$ | **0.893666** | 0.884095 | 0.887945 | 0.8882271 | 0.890979 | 0.893116 | 0.890028 | 0.887805 |
| | MAE | **0.072141** | 0.076484 | 0.074373 | 0.0743641 | 0.073635 | 0.073320 | 0.073715 | 0.075201 |
| | MSE | **0.010236** | 0.011157 | 0.010787 | 0.0107594 | 0.010494 | 0.010289 | 0.010586 | 0.010800 |
| | RMSE | **0.101172** | 0.105627 | 0.103858 | 0.1037274 | 0.102443 | 0.101434 | 0.102888 | 0.103923 |
| | IA | **0.972154** | 0.968608 | 0.970433 | 0.9697823 | 0.970688 | 0.971897 | 0.970706 | 0.969697 |

**Note:**
The best results are shown in bold.

As Table 11 demonstrates, the introduces algorithms performed admirably, attaining the best outcomes on overall evaluations as well as two and three step ahead. The original HHO performed marginally better in one step ahead forecasts when considering at the MAE and IA metrics.

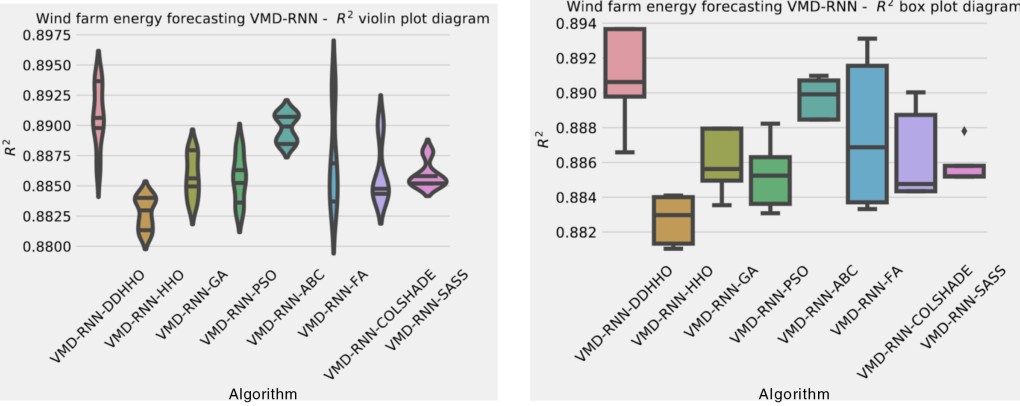

**Figure 10 Wind dataset objective function and $R^2$ distribution plots for each metaheuristic without attention layer.**

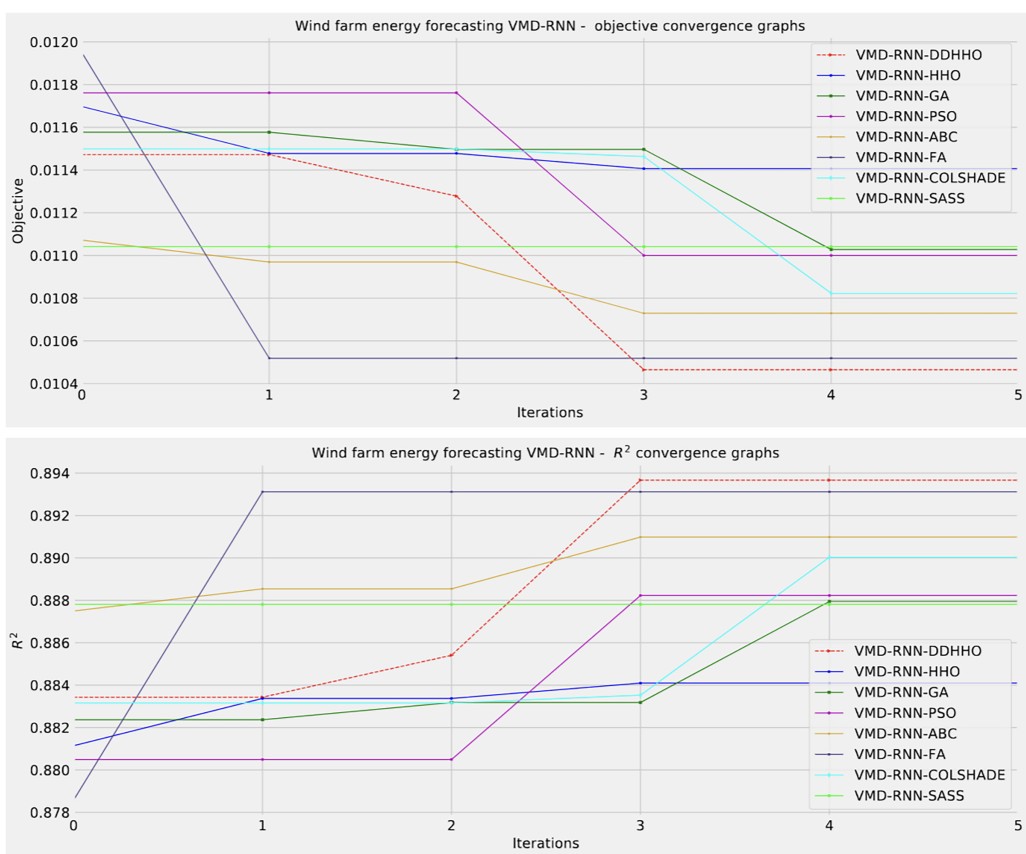

**Figure 11 Wind dataset objective function and $R^2$ convergence plots for each metaheuristic without attention layer.**

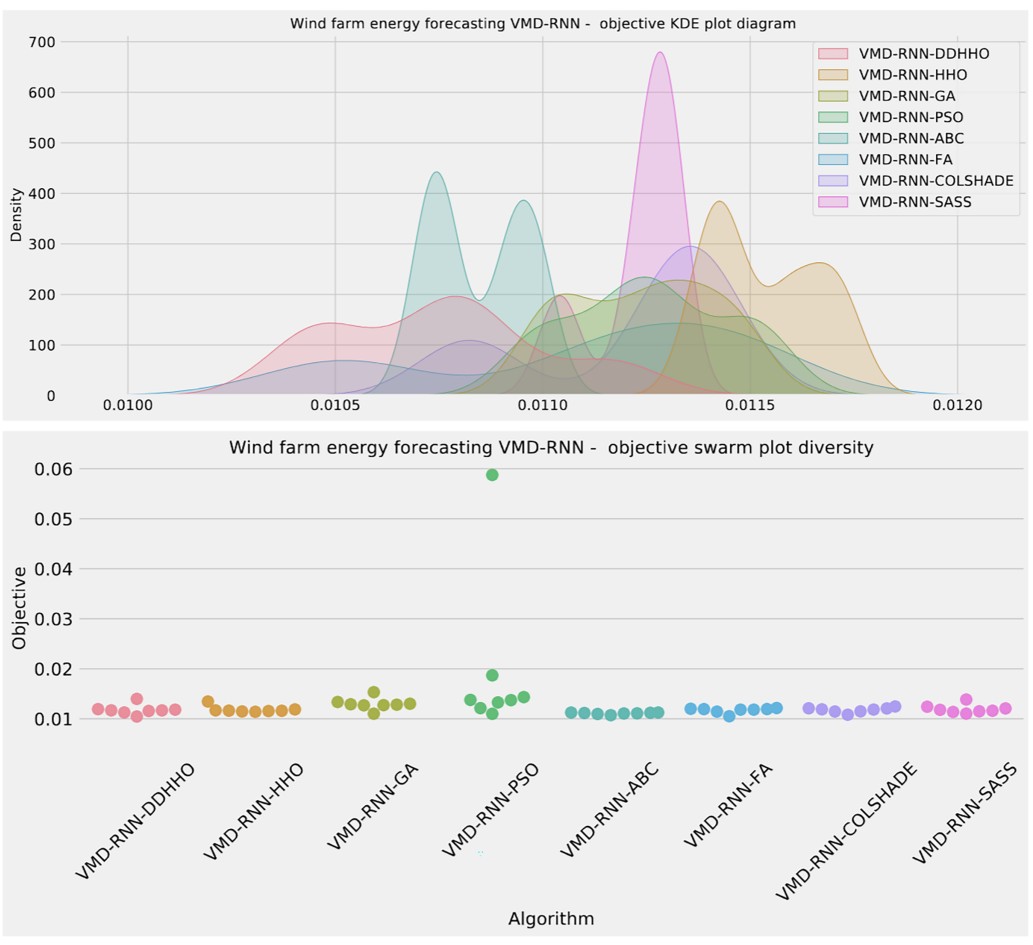

**Figure 12 Wind dataset objective swarm and KDE plots for each metaheuristic without attention layer.**

**Table 9 Parameters for best performing wind prediction RNN model optimized by each metaheuristic.**

| Method | Learning rate | Dropout | Epochs | Layers | L1 Neurons | L2 Neurons | L3 Neurons |
|---|---|---|---|---|---|---|---|
| VMD-RNN-DDHHO | 0.010000 | 0.050755 | 300 | 3 | 97 | 94 | 100 |
| VMD-RNN-HHO | 0.006340 | 0.100000 | 200 | 1 | 100 | / | / |
| VMD-RNN-GA | 0.009989 | 0.067669 | 134 | 2 | 95 | 58 | / |
| VMD-RNN-PSO | 0.008124 | 0.053596 | 294 | 3 | 85 | 93 | 73 |
| VMD-RNN-ABC | 0.010000 | 0.100000 | 300 | 3 | 100 | 79 | 50 |
| VMD-RNN-FA | 0.010000 | 0.050000 | 300 | 2 | 100 | 50 | / |
| VMD-RNN-COLSHADE | 0.010000 | 0.096306 | 300 | 3 | 67 | 50 | 50 |
| VMD-RNN-SASS | 0.010000 | 0.050000 | 300 | 1 | 64 | / | / |

**Table 10  VMD-RNN-ATT wind energy forecasting objective function overall outcomes.**

| Method | Best | Worst | Mean | Median | Std | Var |
|---|---|---|---|---|---|---|
| VMD-RNN-ATT-DDHHO | **0.010359** | **0.011446** | **0.010993** | 0.011361 | 0.000475 | 2.254891E-7 |
| VMD-RNN-ATT-HHO | 0.010806 | 0.011496 | 0.011261 | 0.011424 | 0.000269 | 7.259626E-8 |
| VMD-RNN-ATT-GA | 0.011264 | 0.011672 | 0.011441 | 0.011387 | 0.000152 | 2.298042E-8 |
| VMD-RNN-ATT-PSO | 0.011167 | 0.011808 | 0.011455 | 0.011431 | 0.000251 | 6.293247E-8 |
| VMD-RNN-ATT-ABC | 0.010911 | 0.011524 | 0.011279 | **0.011259** | 0.000220 | 4.861609E-8 |
| VMD-RNN-ATT-FA | 0.011160 | 0.011554 | 0.011360 | 0.011420 | 0.000145 | 2.108468E-8 |
| VMD-RNN-ATT-COLSHADE | 0.011054 | 0.011368 | 0.011203 | 0.011184 | **0.000126** | 1.582216E-8 |
| VMD-RNN-ATT-SASS | 0.011269 | 0.011519 | 0.011392 | 0.011400 | 0.000096 | **9.213128E-9** |

**Note:**
The best results are shown in bold.

**Table 11  The VMD-RNN-ATT wind energy metrics per each step.**

| Step | Metric | VMD-RNN-ATT-DDHHO | VMD-RNN-ATT-HHO | VMD-RNN-ATT-GA | VMD-RNN-ATT-PSO | VMD-RNN-ATT-ABC | VMD-RNN-ATT-FA | VMD-RNN-ATT-COLSHADE | VMD-RNN-ATT-SASS |
|---|---|---|---|---|---|---|---|---|---|
| One step | $R^2$ | **0.869388** | 0.868300 | 0.863840 | 0.860679 | 0.861597 | 0.854800 | 0.860994 | 0.853326 |
| | MAE | 0.080227 | **0.079741** | 0.081451 | 0.083636 | 0.081330 | 0.083773 | 0.082541 | 0.083572 |
| | MSE | **0.012573** | 0.012678 | 0.013107 | 0.013411 | 0.013323 | 0.013977 | 0.013381 | 0.014119 |
| | RMSE | **0.112129** | 0.112595 | 0.114485 | 0.115806 | 0.115425 | 0.118225 | 0.115676 | 0.118823 |
| | IA | 0.964787 | **0.965400** | 0.963486 | 0.963898 | 0.963680 | 0.961305 | 0.963349 | 0.960917 |
| Two step | $R^2$ | **0.902255** | 0.898536 | 0.892452 | 0.895950 | 0.897634 | 0.898030 | 0.897528 | 0.895859 |
| | MAE | **0.070517** | 0.071214 | 0.073747 | 0.073326 | 0.071518 | 0.071795 | 0.072607 | 0.073126 |
| | MSE | **0.009409** | 0.009767 | 0.010353 | 0.010016 | 0.009854 | 0.009816 | 0.009864 | 0.010025 |
| | RMSE | **0.097000** | 0.098828 | 0.101748 | 0.100080 | 0.099267 | 0.099074 | 0.099318 | 0.100124 |
| | IA | **0.973859** | 0.973364 | 0.971348 | 0.972700 | 0.973169 | 0.973293 | 0.973173 | 0.972177 |
| Three step | $R^2$ | **0.912571** | 0.903750 | 0.900340 | 0.902971 | 0.908152 | 0.906962 | 0.904508 | 0.907307 |
| | MAE | **0.067887** | 0.070822 | 0.072048 | 0.071218 | 0.069180 | 0.070399 | 0.072522 | 0.071352 |
| | MSE | **0.008416** | 0.009265 | 0.009593 | 0.009340 | 0.008841 | 0.008956 | 0.009192 | 0.008923 |
| | RMSE | **0.091739** | 0.096255 | 0.097946 | 0.096644 | 0.094028 | 0.094636 | 0.095876 | 0.094460 |
| | IA | **0.976584** | 0.974331 | 0.973022 | 0.973790 | 0.975383 | 0.975599 | 0.974773 | 0.975041 |
| Overall | $R^2$ | **0.894738** | 0.890195 | 0.885544 | 0.886533 | 0.889128 | 0.886597 | 0.887677 | 0.885497 |
| | MAE | **0.0728767** | 0.073925 | 0.075749 | 0.076060 | 0.074010 | 0.075322 | 0.075890 | 0.076017 |
| | MSE | **0.0101326** | 0.010570 | 0.011018 | 0.010922 | 0.010673 | 0.010916 | 0.010812 | 0.011022 |
| | RMSE | **0.1006610** | 0.102810 | 0.104965 | 0.104510 | 0.103309 | 0.104481 | 0.103982 | 0.104986 |
| | IA | **0.9717431** | 0.971032 | 0.969285 | 0.970130 | 0.970744 | 0.970066 | 0.970432 | 0.969378 |

**Note:**
The best results are shown in bold.

Further distribution and convergence graphs for the objective function and $R^2$ are shown in Figs. 13 and 14. Accompanying KDE and swarm diversity plots are given in Fig. 15.

Finally, the selected parameter for the best performing models optimized by each metaheuristic are shown in Table 12.

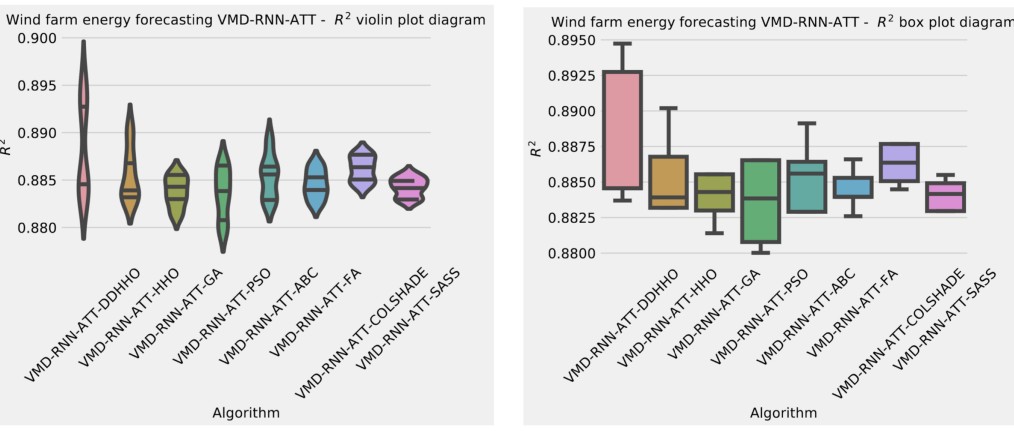

**Figure 13 Wind dataset objective function and $R^2$ distribution plots for each metaheurstic with attention layer.**

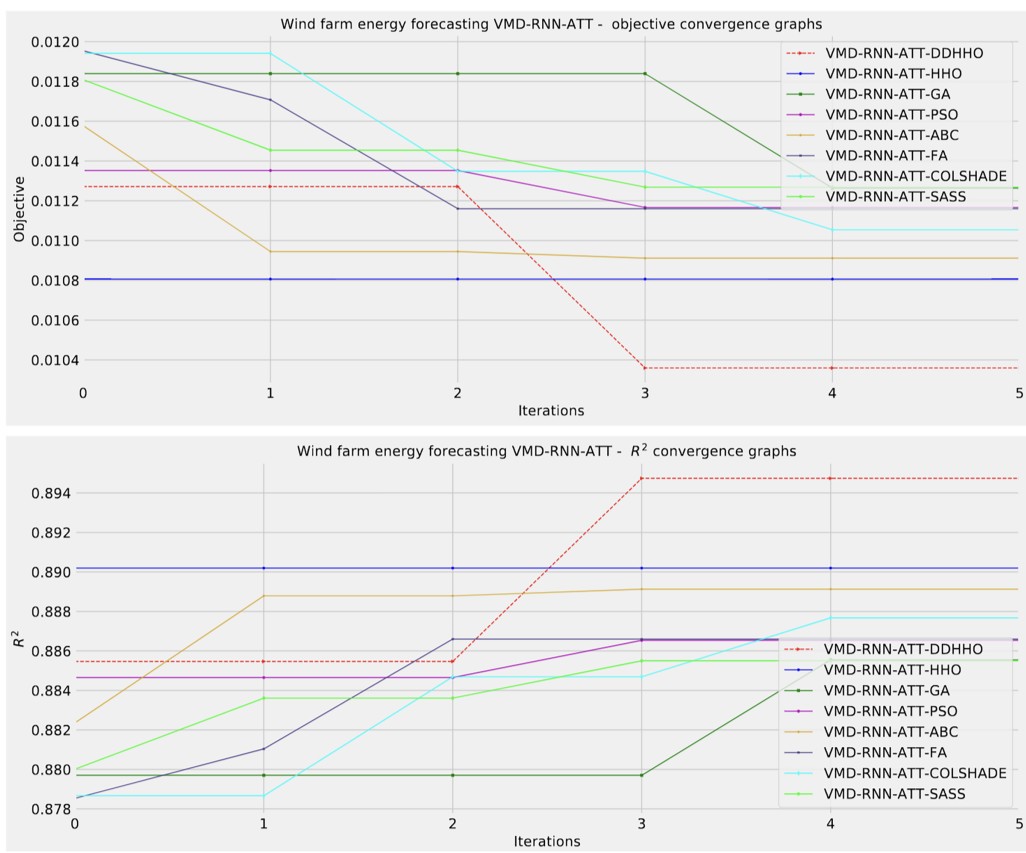

**Figure 14 Wind dataset objective function and $R^2$ convergence plots for each metaheuristic with attention layer.**

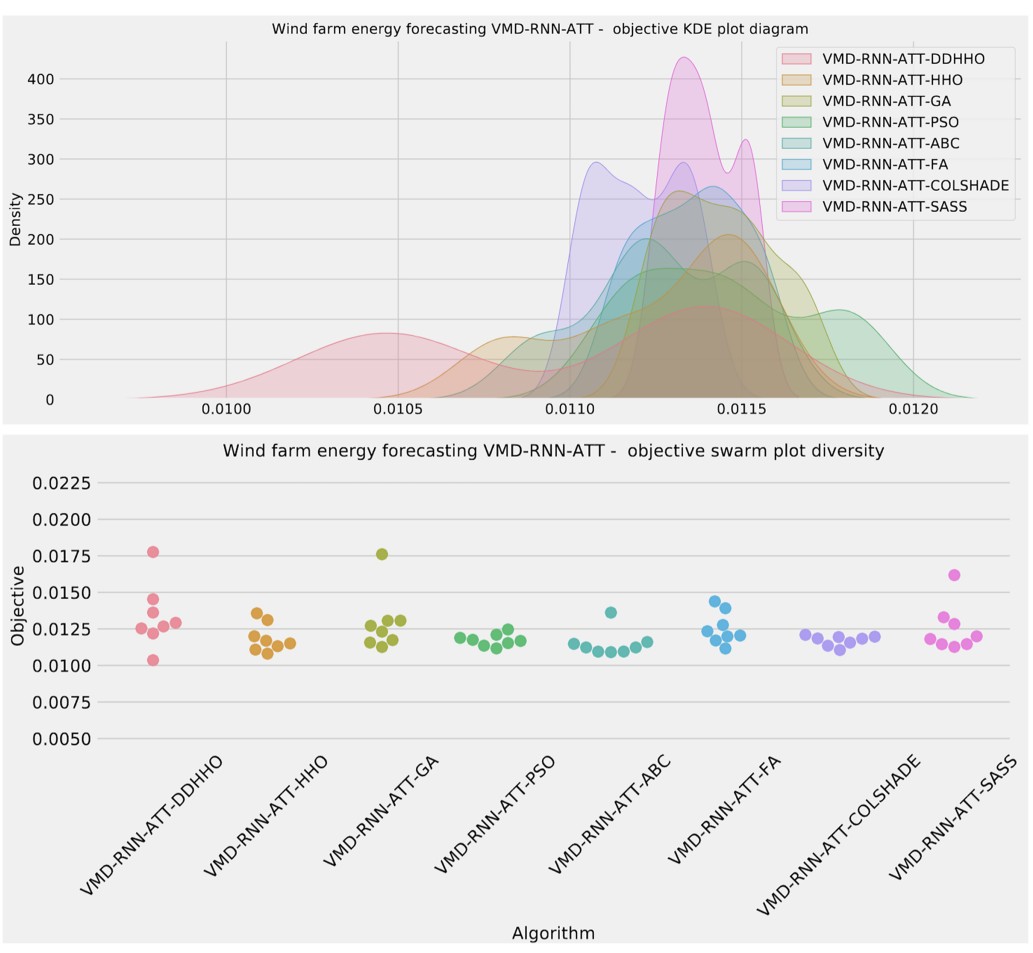

**Figure 15 Wind dataset objective swarm and KDE plots for each metaheuristic with attention layer.**

**Table 12 Parameters for best-performing wind prediction RNN-ATT model optimized by each metaheuristic.**

| Method | Learning rate | Dropout | Epochs | Layers | L1 Neurons | L2 Neurons | L3 Neurons | ATT Neurons |
|---|---|---|---|---|---|---|---|---|
| VMD-RNN-DDHHO | 0.010000 | 0.063597 | 267 | 3 | 69 | 100 | 50 | 77 |
| VMD-RNN-HHO | 0.010000 | 0.100000 | 222 | 1 | 74 | / | / | 54 |
| VMD-RNN-GA | 0.007046 | 0.060227 | 120 | 2 | 66 | 73 | / | 74 |
| VMD-RNN-PSO | 0.010000 | 0.050000 | 234 | 3 | 100 | 50 | 100 | 50 |
| VMD-RNN-ABC | 0.010000 | 0.100000 | 300 | 3 | 100 | 50 | 50 | 50 |
| VMD-RNN-FA | 0.010000 | 0.050000 | 300 | 3 | 50 | 100 | 81 | 98 |
| VMD-RNN-COLSHADE | 0.005840 | 0.100000 | 300 | 1 | 91 | / | / | 86 |
| VMD-RNN-SASS | 0.009995 | 0.100000 | 255 | 1 | 60 | / | / | 100 |

# DISCUSSION, STATISTICAL VALIDATION AND INTERPRETATION

This section presents a discussion of the advantages of the techniques employed in the conducted research, as well as the statistical analysis of the methods used for comparisons, and the interpretation of the best models generated for both datasets.

## Benefits of using attention mechanism for renewable power generation forecasting

The attention mechanism has emerged as a powerful tool in the field of machine learning, particularly for sequence-to-sequence learning problems like renewable power generation forecasting. By selectively focusing on different parts of the input sequence when generating the output, the attention mechanism can enhance the performance of forecasting models like the Luong attention-based RNN model. Below, we discuss the key benefits of using attention mechanisms for renewable power generation forecasting:

**1. Improved long-term dependency handling:** Renewable power generation data often exhibit long-term dependencies due to factors like seasonal patterns and weather trends. Traditional RNN models can struggle to capture these long-term dependencies effectively, leading to suboptimal forecasts. The mechanism of attention introduces different importance weights for seperate input sequence parts, enabling it to focus on the most relevant information for generating the output, thus better handling long-term dependencies.

**2. Enhanced forecasting accuracy:** The attention mechanism can lead to more accurate forecasts by enabling the model to focus on the most relevant parts of the input sequence when generating the output. This selective focus allows the model to capture the underlying patterns and relationships within the renewable power generation data more effectively, resulting in improved forecasting performance.

**3. Interpretability:** Attention mechanisms provide a level of interpretability to the model's predictions by highlighting which parts of the input sequence have the most significant impact on the output. This interpretability can be particularly valuable in renewable power generation forecasting, as it allows domain experts to gain insights into the factors influencing the model's forecasts and to validate the model's predictions based on their domain knowledge.

**4. Robustness to noise and irrelevant information:** Renewable power generation data can be subject to noise and irrelevant information (*e.g.*, due to measurement errors or unrelated external factors). The attention mechanism can help in mitigating the impact of such disturbances on the model's forecasts by selectively focusing on the most relevant parts of the input sequence and down-weighting the influence of noise and irrelevant information.

**5. Scalability:** Attention mechanisms can scale well with large input sequences, as they allow the model to focus on the most relevant parts of the input sequence without the need to process the entire sequence in a fixed-size hidden state. This scalability can be particularly beneficial for renewable power generation forecasting problems, where the

input data may consist of long sequences of historical power generation measurements and environmental variables.

**6. Flexibility:** Attention mechanisms can be easily incorporated into various RNN architectures, such as LSTM and GRU, providing flexibility in designing and adapting the forecasting model for different renewable power generation scenarios and data characteristics.

An additional note needs to be made on attention mechanisms. The attained results suggest that networks utilizing the attention mechanisms perform slightly worse then the basic RNN. This is likely due to networks with attention layers having a deeper network architecture and thus require more training epochs to improve performance.

## Benefits of time series decomposition and integration

Incorporating time-series decomposition and integration into the Luong attention-based RNN model can offer several benefits for renewable power generation forecasting:

**1. Improved forecasting accuracy:** By decomposing the time-series and accounting for its components, the model can better capture the underlying patterns and dependencies in the data, potentially leading to more accurate and reliable forecasts.

**2. Enhanced model interpretability:** Decomposition provides insights into the different components of the time-series, making it easier to understand and interpret the model's predictions in terms of trend, seasonality, and residual components.

**3. Robustness to noise:** By separating the noise component from the trend and seasonal components, the decomposition process can help in reducing the impact of noise and outliers on the model's forecasts, making the model more robust to disturbances.

**4. Flexibility and customizability:** Decomposition and integration techniques can be adapted and fine-tuned to suit the specific characteristics and requirements of the renewable power generation data, allowing for a more flexible and customizable forecasting approach.

**5. Improved model performance:** The integration of decomposed components into the RNN model can help in better capturing the relationships between the components and the target variable, potentially leading to improved model performance in terms of generalization and predictive accuracy.

## Statistical analysis

When considering optimization problems, assessing models is an important topic. Understanding the statistical significance of the introduced enhancements is crucial. Outcomes alone are not adequate to state that one algorithms is superior to another one. Previous research suggests (*Derrac et al., 2011*) that a statistical assessment should take place only after the methods being evaluated are adequately sampled. This is done by ascertaining objective averages over several independent runs. Additionally, samples need to originate form a normal distribution so as to avoid misleading conclusions. The use of objective function averages is still for comparison of stochastic methods is still an open question among researchers (*Eftimov, Korošec & Seljak, 2017*). To ascertain statistical significance of the observed outcomes the best values over 30 independent executions of

**Table 13 Shapiro-Wilk scores for the single-problem analysis for testing normality condition.**

| Experiment | DDHHO | HHO | GA | PSO | ABC | FA | COLSHADE | SASS |
|---|---|---|---|---|---|---|---|---|
| Solar VMD-RNN | 0.035 | 0.023 | 0.022 | 0.026 | 0.027 | 0.030 | 0.017 | 0.014 |
| Solar VMD-RNN-ATT | 0.035 | 0.032 | 0.037 | 0.019 | 0.022 | 0.025 | 0.037 | 0.033 |
| Wind VMD-RNN | 0.029 | 0.020 | 0.025 | 0.036 | 0.033 | 0.019 | 0.026 | 0.024 |
| Wind VMD-RNN-ATT | 0.021 | 0.028 | 0.025 | 0.037 | 0.035 | 0.024 | 0.026 | 0.041 |

**Table 14 Wilcoxon signed-rank test findings.**

| DDHHO *vs* others | HHO | GA | PSO | ABC | FA | COLSHADE | SASS |
|---|---|---|---|---|---|---|---|
| Solar VMD-RNN | 0.035 | 0.046 | 0.036 | **0.062** | 0.043 | 0.029 | 0.040 |
| Solar VMD-RNN-ATT | 0.041 | 0.044 | 0.046 | 0.035 | 0.024 | 0.039 | 0.037 |
| Wind VMD-RNN | 0.024 | 0.043 | 0.039 | **0.052** | 0.045 | 0.044 | 0.038 |
| Wind VMD-RNN-ATT | 0.039 | 0.027 | 0.025 | 0.038 | 0.035 | 0.042 | 0.032 |

**Note:**
The best results are shown in bold.

each metaheuristic have been used for creating the samples. However, the safe use of parametric tests needed to be confirmed. For this, independence, normality, and homoscedasticity of the data variances were considered as recommended by *LaTorre et al. (2021)*. The independence criterion is fulfilled due to the fact that each run is initialized with an pseudo-random number seed. However, the normality condition is not satisfied as the obtained samples do not stem from a normal distribution as shown by the KED plots and proved by the Shapiro-Wilk test outcomes for single-problem analysts (*Shapiro & Francia, 1972*). By performing the Shapiro-Wilk test, $p$-values are generated for each method-problem combination, and these outcomes are presented in Table 13.

The standard significance levels of $\alpha = 0.05$ and $\alpha = 0.1$ suggest that the null hypothesis ($H0$) can be refuted, which implies that none of the samples (for any problem-method combinations) are drawn from a normal distribution. This indicates that the assumption of normality, which is necessary for the reliable use of parametric tests, was not satisfied, and therefore, it was deemed unnecessary to verify the homogeneity of variances.

As the requirements for the reliable application of parametric tests were not met, non-parametric tests were employed for the statistical analysis. Specifically, the Wilcoxon signed-rank test, which is a non-parametric statistical test (*Taheri & Hesamian, 2013*), was performed on the DDHHO method and all other techniques for all three problem instances (experiments). The same data samples used in the previous normality test (Shapiro-Wilk) were used for each method. The results of this analysis are presented in Table 14, where $p$-values greater than the significance level of $\alpha = 0.05$ are highlighted in bold.

Table 14, which presents the $p$-values obtained from the Wilcoxon signed-rank test, demonstrate that, except for the ABC algorithm in the experiment where VMD-RNN was optimized and validated against solar and wind datasets, the proposed DDHHO method achieved significantly better performance than all other techniques in all three experiments. When compared with ABC, the calculated $p$-value was slightly above the 0.05 threshold

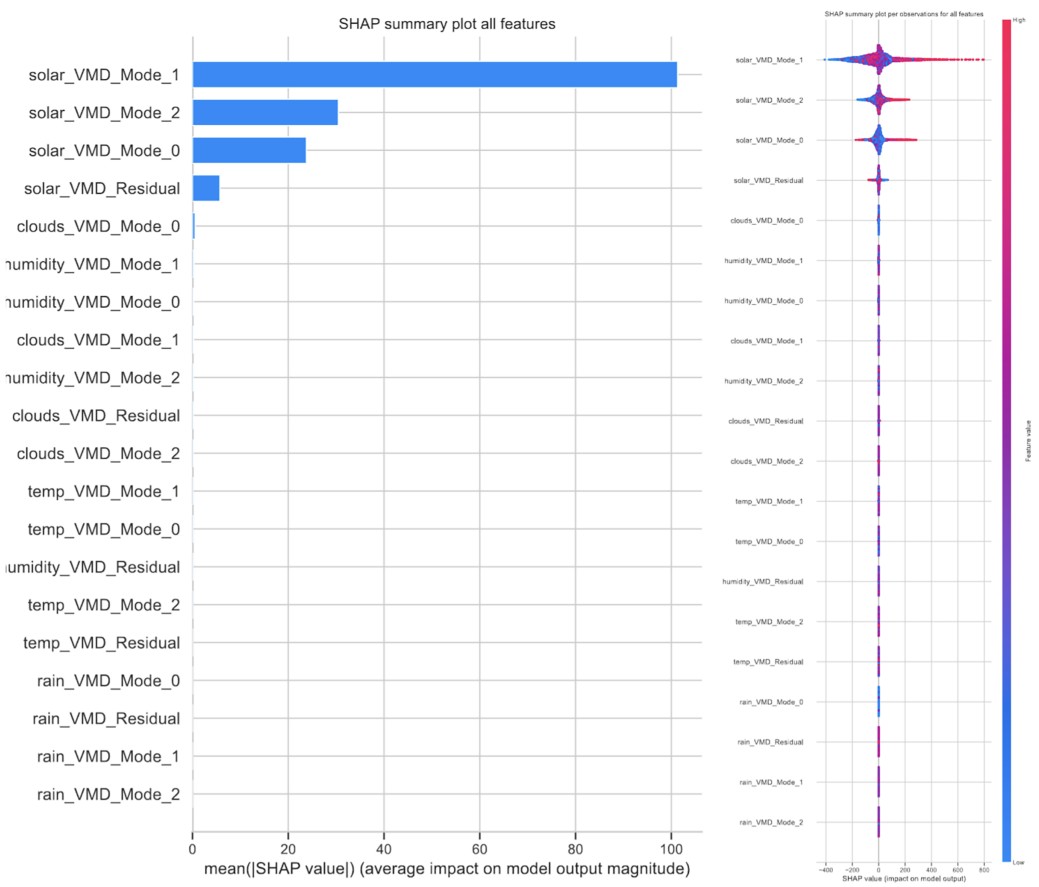

**Figure 16 Feature impacts for the best performing RNN model for solar forecasting.**

(highlighted in bold in Table 14), suggesting that the DDHHO performed comparably to ABC. This was expected for the solar dataset, since the ABC in this simulation achieved moderately better mean value than the DDHHO, as demonstrated in Table 1.

The *p*-values for all other methods were lower than 0.05. Therefore, the DDHHO technique exhibited both robustness and effectiveness as an optimizer in these computationally intensive simulations. Based on the statistical analysis, it can be concluded that the DDHHO method outperformed most of the other metaheuristics investigated in all four experiments.

## Best model interpretation and feature importance

SHAP (*Lundberg & Lee, 2017*) is a method that can be utilized to interpret the outputs of various AI models. Game theory provides a strong basis for SHAP. Though the use of SHAP the influence real-world factors have on model predictions can be determined. In order to determine the factors that play the highest role in energy production in solar and wind generation the best models with the highest performance output have been subjected to analysis. The outcomes for solar generation are shown in Fig. 16, while wind generation is shown in Fig. 17.

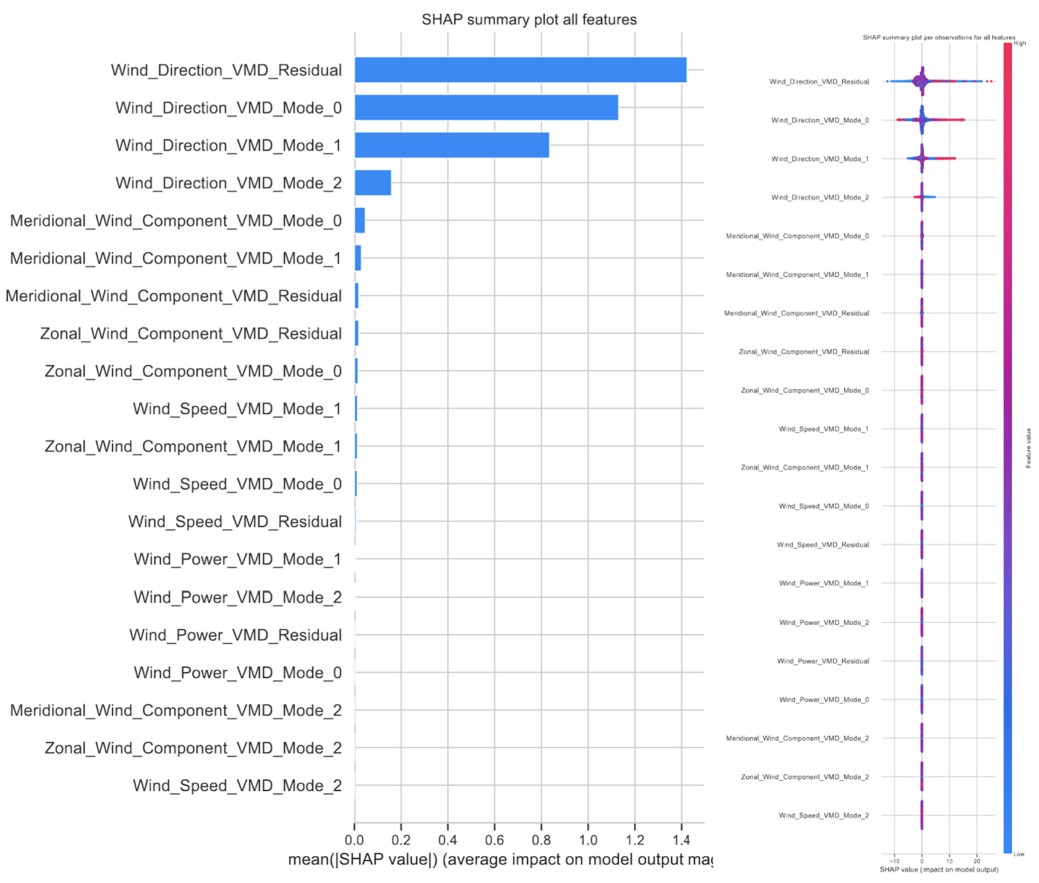

**Figure 17 Feature impacts for the best performing RNN model for wind forecasting.**

As demonstrated by Fig. 16, a significant influence of previous solar generation instances can be observed. Cloud cover and humidity play a minor role in forecasting, with cloud cover decreasing the power produced by the photovoltaic cells.

Indicators form Fig. 17 suggest that when forecasting wind power generation wind direction modes have an important role. However, likely due to the sporadic nature of wind bursts wind generation residuals have the highest impact on predictions. Finally, the meridional followed by zonal wind components pay a minor role in forecasting.

## CONCLUSIONS

This study presents a novel attention-based recurrent neural network model for multistep ahead time-series forecasting of renewable power generation, demonstrating improved forecasting accuracy on both Spain's wind and solar energy datasets and China's wind farm dataset. The HHO algorithm is employed for hyperparameter optimization, addressing the challenges posed by the large number of hyperparameters in RNN-type networks. The attention model applied in the second group of experiments provides a weighting system to the RNN, further enhancing the model's performance. The proposed approach has the potential to significantly impact the transition towards a more

sustainable future by addressing key challenges related to the storage and management of renewable power generation.

As with any work this research has several limitations. Other methods exist for tackling time-series forecasting and their potential remains yet to be explored. Further potential for improvement exist for the HHO, as well as other metaheuristic algorithms yet to be applied to cloud forecasting. Additionally, other approaches for interpreting feature influence exist such as through the analysis of n-Shapley values.

Future research will focus on refining the HHO algorithm for hyperparameter optimization and exploring additional decomposition methods to further improve the forecasting capabilities of the proposed approach, as well as exploring additional metaheuristics applied to clout load forecasting. Additionally, further methods for feature impact interpretation will be explored.

### Funding
The authors received no funding for this work.

### Competing Interests
Robertas Damaševičius is an Academic Editor for PeerJ.

### Author Contributions
- Robertas Damaševičius analyzed the data, authored or reviewed drafts of the article, and approved the final draft.
- Luka Jovanovic conceived and designed the experiments, performed the experiments, analyzed the data, performed the computation work, prepared figures and/or tables, and approved the final draft.
- Aleksandar Petrovic conceived and designed the experiments, performed the experiments, analyzed the data, performed the computation work, prepared figures and/or tables, and approved the final draft.
- Miodrag Zivkovic conceived and designed the experiments, performed the experiments, analyzed the data, performed the computation work, prepared figures and/or tables, and approved the final draft.
- Nebojsa Bacanin conceived and designed the experiments, performed the experiments, analyzed the data, prepared figures and/or tables, authored or reviewed drafts of the article, and approved the final draft.
- Dejan Jovanovic conceived and designed the experiments, performed the experiments, analyzed the data, performed the computation work, prepared figures and/or tables, and approved the final draft.
- Milos Antonijevic conceived and designed the experiments, performed the experiments, analyzed the data, performed the computation work, prepared figures and/or tables, and approved the final draft.

## Data Availability

The code is available in the Supplemental File.

The data is available at Zenodo: Robertas Domasevicius, Luka Jovanovic, Aleksandar Petrovic, Miodrag Zivkovic, Nebojsa Bacanin, Dejan Jovanovic, & Milos Antonijevic. (2023). Decomposition aided attention-based recurrent neural networks for multistep ahead time-series forecasting of renewable power generation. https://zenodo.org/records/7964916.

## Supplemental Information

Supplemental information for this article can be found online at http://dx.doi.org/10.7717/peerj-cs.1795#supplemental-information.

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
