# Peer review of "Decomposition aided attention-based recurrent neural networks for multistep ahead time-series forecasting of renewable power generation"

_PeerJ Computer Science, doi:10.7717/peerj-cs.1795_

## Round 0.1 · original submission · Major Revisions

Four reviewers made many valuable comments. The authors are requested to revise it carefully.

**Language Note:** The review process has identified that the English language must be improved. PeerJ can provide language editing services - please contact us at copyediting@peerj.com for pricing (be sure to provide your manuscript number and title). Alternatively, you should make your own arrangements to improve the language quality and provide details in your response letter. – PeerJ Staff

Reviewer 1 ·

Basic reporting

Language score below par; needs thorough proofread.
Old work cited; need to include state-of-art work.

Experimental design

Needs rigorous investigation.
Empirical results must be presented in an appropriate manner.

Validity of the findings

Unable to validate practical feasibility.

Reviewer 2 ·

Basic reporting

This paper (ID: peerj-reviewing-86288) entitled “Decomposition aided attention-based recurrent neural networks for multistep ahead time-series forecasting of renewable power generation”, this study proposes an attention-based recurrent neural network approach for forecasting power generated from renewable sources. And a modified metaheuristic is introduced to optimized hyper parameter values of the utilized networks. The model developed was applied to two datasets and the model's capability to fit and generalize was tested. Finally, the best-performing model was interpreted using Shapley Additive exPlanations. This work helps to improve the understanding of renewable energy generation forecasting and the model building optimization process can be used as a reference for other scholars' research. Several issues should be addressed before recommending it to publish in PeerJ Computer Science.
1. In the second part of the manuscript, a great deal of terminology is introduced that could be merged with the hyper parameter regulation section to streamline the language in that section. Show more of the authors' research.
2. The fourth part of the manuscript applies the model to two datasets, selecting some of the influences and the generated photovoltaic power as inputs to the model as well as target features, respectively. But the generated photovoltaic power has many influencing factors, please clarify the reasons for choosing influencing factors such as humidity, rainfall, cloud cover and ambient temperature.
3. The font formatting of the tables in the text needs to be optimized (e.g., font sizes need to be adjusted in Tables 2 and 5), and it is recommended that the full text be checked for optimization.
4. Punctuation is used incorrectly and it is recommended that the whole article be checked and corrected. (For example, in the manuscript, there is an extra space before the colon in equation (10).)
5. It is suggested that more model hyper parameter optimization processes could be added to the fifth part of the manuscript.

Experimental design

see above

Validity of the findings

see above

Additional comments

see above

·

Basic reporting

1. The article is written clearly and professionally.
2. Sufficient background are presented from predictive models and feature engineering.
3. The article is well structured.
4. The results are solid with lots of experiments.
5. Tables and figures are clear.

Experimental design

1. For the metric selection, why don't authors consider the normlized metric like MAPE?

2. One of the main contribution of this paper is the HHO algorithm for the hyper-parameters tuning. Also, Bayesion optimization is a good choice for benchmark, I'd like to see the results for VMD-RNN-BO.

3. Could you also compare the complexity of those algorithms and how your enchanced HHO is well placed?

Validity of the findings

1. Performance running under heuristic algorithms could be relatively random. Instead of reporting the number solely, reporting mean and variance under several tests (like 5 or 10) is more robust.

2. Regarding several tables, three steps PSO seems better than other tuning algorithm, including HHO. Any explaination for that? What if we try more steps?

3.Regarding the feature importance, just first 4 of them are important (counting more than 95%). What if we just keep those most important features? Would the performance keep similar?

Additional comments

Well structured and elaborated stroy telling, please try to answer those questions then we can make the manuscript more strong.

Reviewer 4 ·

Basic reporting

General Comments
Language and Formatting: It seems that the paper has some typos and unconventional characters. Please review the paper thoroughly to ensure a high standard of presentation. For example, "utilized applied the time series" in the abstract seems awkwardly phrased.

Methodology: Original Harris Hawk optimization (HHO)
Motivation: The introduction to HHO and its modification needs more context for the reader to understand its relevance. Why was this specific algorithm chosen?

Mathematical Notations: Equations are generally well presented, but make sure that all variables are clearly defined when first introduced. For instance, it's not quite clear what
E0 stands for in Equation (6).

Methodology: Proposed Enhanced HHO
New Initialization Scheme: While a new initialization strategy is presented, it is not entirely clear how it substantially benefits over the traditional methods. More justification or empirical evidence would help.

Data Preprocessing
Missing Data Imputation: What imputation methods were considered and why? This could affect model performance significantly.

Data Splitting: It's good that the data split proportions are mentioned. But, is there any temporal aspect considered during the split? For example, using the last N months for testing.

Feature Engineering: While additional features like moving averages are mentioned, their inclusion or exclusion's empirical effect is not discussed. It would be useful to include such insights.

Normalization: What scaling technique was used? Min-Max scaling or Z-score normalization?

Experimental Setup
Datasets: Given that the paper is about renewable energy, it might be worth mentioning more about the datasets. Are they publicly available? What is the time period covered?

Model Comparison: While you mentioned that the models are compared with state-of-the-art optimizers, it would be beneficial to list which ones were considered for benchmarking.

Validation: Could you explain how model robustness is validated? For example, are any sensitivity analyses conducted?

Please consider these suggestions to improve the manuscript. I look forward to seeing the revised version.

Experimental design

No comment

Validity of the findings

No comment

Additional comments

No comment

---

## Round 0.2 · accepted · Accept

I am writing to inform you that your manuscript has been Accepted for publication. Congratulations!

Reviewer 2 ·

Basic reporting

All my concerned issues were addressed.

Experimental design

All my concerned issues were addressed.

Validity of the findings

All my concerned issues were addressed.

Additional comments

All my concerned issues were addressed.

·

Basic reporting

The revised manuscript is clear and unambiguous. Regarding my concerns, authors addressed them either by updating the manuscript or explaining details in the rebuttlal. I don't have any questions and this manuscrip should be accepted.

Experimental design

See 1.

Validity of the findings

See 1.

Reviewer 4 ·

Basic reporting

The authors have addressed all my concerns.

Experimental design

The authors have addressed all my concerns.

Validity of the findings

The authors have addressed all my concerns.